# Eosinophils preserve bone homeostasis by inhibiting excessive osteoclast formation and activity via eosinophil peroxidase

Darja Andreev[1,2] ✉, Katerina Kachler[1,2], Mengdan Liu[1,2,3], Zhu Chen[1,2,4], Brenda Krishnacoumar[1,2,5], Mark Ringer[1,2], Silke Frey[1,2], Gerhard Krönke ®[1,2,6], David Voehringer ®[7], Georg Schett ®[1,2] & Aline Bozec ®[1,2] ✉

Eosinophils are involved in tissue homeostasis. Herein, we unveiled eosinophils as important regulators of bone homeostasis. Eosinophils are localized in proximity to bone-resorbing osteoclasts in the bone marrow. The absence of eosinophils in ΔdblGATA mice results in lower bone mass under steady-state conditions and amplified bone loss upon sex hormone deprivation and inflammatory arthritis. Conversely, increased numbers of eosinophils in IL-5 transgenic mice enhance bone mass under steady-state conditions and protect from hormone- and inflammation- mediated bone loss. Eosinophils strongly inhibit the differentiation and demineralization activity of osteoclasts and lead to profound changes in the transcriptional profile of osteoclasts. This osteoclast-suppressive effect of eosinophils is based on the release of eosinophil peroxidase causing impaired reactive oxygen species and mitogen-activated protein kinase induction in osteoclast precursors. In humans, the number and the activity of eosinophils correlates with bone mass in healthy participants and rheumatoid arthritis patients. Taken together, experimental and human data indicate a regulatory function of eosinophils on bone.

Bone is continuously remodeled throughout life. Up to 10% of the calcified bone is renewed every year, which depends on a fine balance between bone resorption and bone formation. Osteoclasts are critical cells for bone homeostasis as they are the only bone-resorbing cells in the body. Increased osteoclast function results in pathological bone loss leading to osteoporosis, defined by low bone mass and increased risk of fracture[1]. Osteoclast activity therefore needs to be tightly controlled by inhibitory cells and pathways.

Osteoclasts are polynucleated cells that are formed by the fusion of precursor cells from the monocyte/macrophage lineage[2,3]. This process requires the presence of osteoclastogenic cytokines, including macrophage colony-stimulating factor (M-CSF) and receptor activator of nuclear factor kappa-B ligand (RANKL)[4,5]. The interaction of RANKL with its receptor RANK on osteoclast precursors triggers a downstream signal via the mitogen-activated protein kinases (MAPKs), ERK (extracellular signal-regulated kinase) 1/2, JNK (c-Jun N-terminal kinase) and p38, activating the master regulator of osteoclastogenesis, nuclear factor of activated T cells 1 (NFATc1)[6,7]. This transcription factor thereupon drives the expression of several proteins necessary for proper bone resorption, such

[1]Department of Internal Medicine 3 – Rheumatology and Immunology, Friedrich-Alexander-University (FAU) Erlangen-Nürnberg and Universitätsklinikum Erlangen, Erlangen, Germany. [2]Deutsches Zentrum für Immuntherapie (DZI), Erlangen, Germany. [3]Department of Rheumatology, Zhejiang University – School of Medicine, Hangzhou, China. [4]Department of Rheumatology and Immunology, Anhui Medical University Affiliated Provincial Hospital, Hefei, China. [5]Institute of Physiology, University Hospital Essen, University of Duisburg-Essen, Essen, Germany. [6]Department of Rheumatology and Clinical Immunology, Charité University Medicine, Berlin, Germany. [7]Department of Infection Biology, FAU Erlangen-Nürnberg and Universitätsklinikum Erlangen, Erlangen, Germany. ✉e-mail: darja.andreev@uk-erlangen.de; aline.bozec@uk-erlangen.de

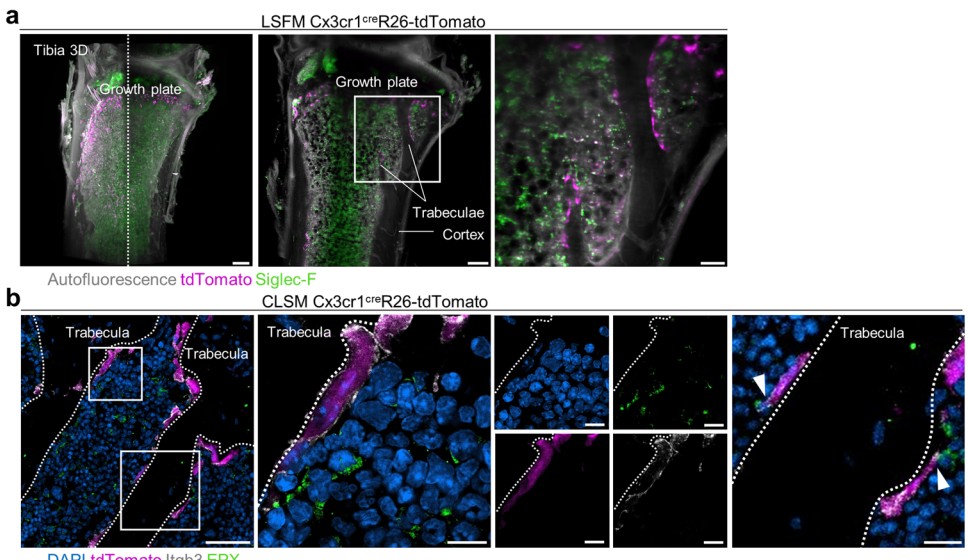

**Fig. 1 | Eosinophils are localized in proximity to bone-resorbing osteoclasts.**
**a** Light sheet fluorescence microscopy (LSFM) of tibial bones from Cx3cr1creR26-tdTomato mice injected with 5 µg anti-Siglec-F AF-647. tdTomato-positive (purple) osteoclasts surround the trabeculae (autofluorescence/gray), Siglec-F-positive (green) eosinophils are in proximity to osteoclasts. Scale bar, 200 µm (left), 150 µm (middle), and 100 µm (right). **b** Confocal laser scanning microscopy (CLSM) of tibial bone sections from Cx3cr1creR26-tdTomato mice. Polynucleated osteoclasts attached to the trabecular bone are marked by tdTomato (purple) and Integrin β-3 (Itgb3/gray), eosinophils are stained with eosinophil peroxidase (EPX/green), and nuclei are visualized by DAPI (blue). Scale bar, 40 µm (left), 7 µm (middle) and 10 µm (right). White arrows point out direct cell-cell interaction between eosinophils and osteoclasts. LSFM and CLSM were performed independently 3 times with 3 mice per experiment.

as tartrate-resistant acid phosphatase (TRAP), matrix metalloproteinase 9 (MMP-9) and cathepsin K (CTSK)[8].

Osteoclasts are controlled by the immune system. In this regard, the pro-osteoclastogenic function of the innate and adaptive immune system has been thoroughly studied. Several cytokines produced by immune cells, such as TNFα, IL-6, IL-1, and IL-17, are highly active in stimulating osteoclast differentiation and function and thus promote immune-mediated bone loss[9]. However, some cytokines, i.e., those related to type 2 immune responses, like IL-4 and IL-13, have been reported to inhibit osteoclast differentiation[10]. Interestingly, these cytokines are highly secreted by eosinophils. To date, the role of eosinophils in bone homeostasis has not been defined.

Eosinophils are part of the innate immune system and belong to the granulocyte lineage. They are formed in the bone marrow, where common myeloid progenitors differentiate into eosinophil progenitors under the influence of the transcription factor, GATA-binding factor 1 (GATA-1). On further maturation, eosinophils express the surface proteins Siglec-F and IL-5 receptor[11,12]. IL-5 is the key cytokine driving eosinophil maturation, activation and survival[13]. A proportion of eosinophils leaves the bone marrow to peripheral sites[14,15]. Usually, eosinophils are associated with the immune defense against parasites by the production of heme peroxidases, such as eosinophil peroxidase (EPX) stored in cytoplasmic granules. While eosinophils participate in diseases that are related to type 2 immune activation, like asthma[16,17], several studies also highlighted the homeostatic functions of eosinophils. The presence of a homeostatic subset of eosinophils has been unveiled in various tissues, such as the bone marrow, the lungs, the intestinal tract, the thymus and the adipose tissue[18]. Recently, we uncovered the presence of a regulatory eosinophil subset in the synovium that promotes the resolution of inflammatory arthritis[19]. Moreover, a 2001 study by Macias et al. identified that constitutive expression of eosinophil-inducing IL-5 results in unique effects on bone metabolism, such as the formation of ectopic bone nodules in the spleen upon age[20].

Based on these observations, we speculated whether eosinophils may represent a homeostatic immune cell lineage in bone remodeling that may prevent overshooting osteoclast formation and bone resorption. Considering that the bulk of eosinophils is localized in the bone marrow, a functional interaction between eosinophils with osteoclasts seems likely. We therefore assessed the functional impact of eosinophils on bone homeostasis with a focus on the regulation of osteoclast function and bone loss.

In this study, we demonstrate that eosinophils play a role in regulating osteoclast activity both in normal bone homeostasis and under pathological conditions. The mechanism involves the secretion of EPX, which leads to the downregulation of reactive oxygen species (ROS) and MAPKs-mediated osteoclastogenesis. Consequently, eosinophils reveal a previously overlooked link between the innate immune system and bone, emphasizing their involvement in preserving tissue homeostasis within the bone compartment.

## Results
### Eosinophils influence steady-state bone remodeling
Eosinophil lineage commitment and differentiation takes place in the bone marrow. Although a certain number of eosinophils migrates into peripheral tissues via the bloodstream, a vast majority of the cells is situated in the bone marrow, where also the bone-resorbing osteoclasts are located[21]. In order to verify a possible interaction of eosinophils with osteoclasts, we performed light sheet fluorescence microscopy (LSFM) of tibial bones from Cx3cr1creR26-tdTomato mice (to visualize osteoclasts) injected with 5 µg anti-Siglec-F AF-647 antibody (to mark eosinophils). The slides demonstrate the presence of eosinophils in both the metaphysis and diaphysis of long bones, suggesting potential interactions with both osteoclast precursors and mature osteoclasts (Fig. 1a). The localization of eosinophils and osteoclasts was further characterized by confocal laser scanning microscopy (CLSM) of tibial bone sections from Cx3cr1creR26-tdTomato mice that were stained with anti-integrin β-3 (Itgb3, for osteoclasts) and with anti-EPX (for eosinophils), illustrating a direct contact between these two cell types (Fig. 1b).

Given the potential interaction between eosinophils and osteoclasts in bone, our objective was to explore the impact of eosinophils

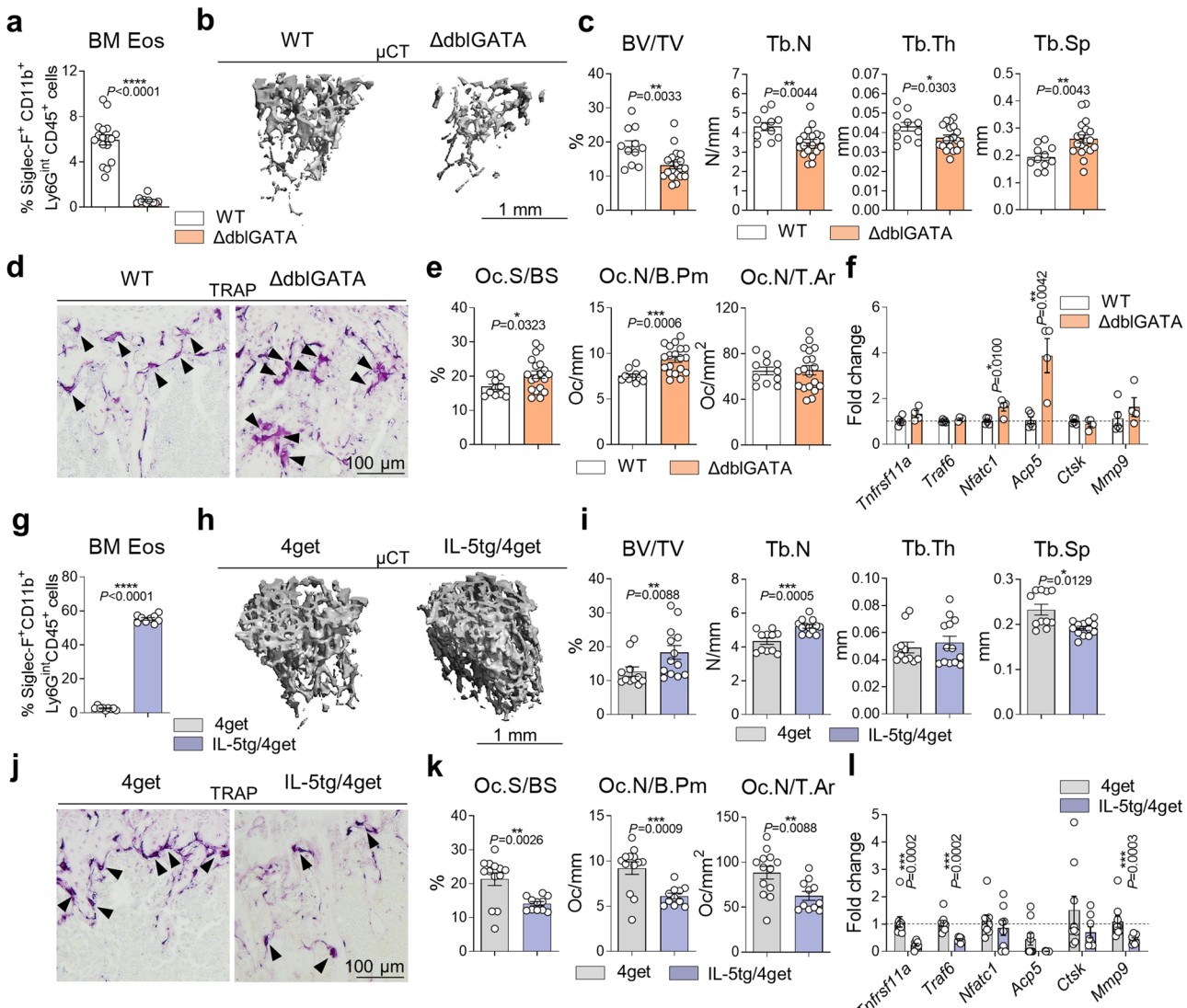

**Fig. 2 | Eosinophils influence steady-state bone remodeling. a** Percentage of eosinophils (Siglec-F⁺CD11b⁺Ly6GᶦⁿᵗCD45⁺ cells) in the bone marrow (BM) of wild type (WT) and ΔdblGATA mice ($n = 17$, 11) analyzed by flow cytometry (FC). **b**, **c** 3D trabecular structure by micro-computed tomography (μCT) (**b**) and quantification of bone volume per total volume (BV/TV), trabecular number (Tb.N), trabecular thickness (Tb.Th), and trabecular separation (Tb.Sp) (**c**) in the tibial bone from WT and ΔdblGATA mice ($n = 11$, 19). Scale bar, 1 mm. **d**, **e** Representative tartrate-resistant acid phosphatase (TRAP) staining (**d**) and quantification of osteoclast surface per bone surface (Oc.S/BS), osteoclast number per bone perimeter (Oc.N/B.Pm), and osteoclast number per tissue area (Oc.N/T.Ar) (**e**) in tibial bone sections from WT and ΔdblGATA mice ($n = 11$, 19). Triangles highlight polynucleated osteoclasts. Scale bar, 100 μm. **f** mRNA expression of *Tnfrsf11a*, *Traf6*, *Nfatc1*, *Acp5*, *Ctsk*, and *Mmp9* in the long bone of WT and ΔdblGATA mice ($n = 5$, 4). **g** Percentage of eosinophils in the BM

of control 4get and IL-5tg/4get mice ($n = 9$) analyzed by FC. **h**, **i** 3D μCT trabecular structure (**h**) and quantification of BV/TV, Tb.N, Tb.Th, and Tb.Sp (**i**) in tibial bone from 4get and IL-5tg/4get mice ($n = 11$, 13). Scale bar, 1 mm. **j**, **k** Representative TRAP staining (**j**) and quantification of Oc.S/BS, Oc.N/B.Pm, and Oc.N/T.Ar (**k**) in tibial bone sections from 4get and IL-5tg/4get mice ($n = 13$, 11). Triangles illustrate polynucleated osteoclasts. Scale bar, 100 μm. **l** mRNA expression of *Tnfrsf11a*, *Traf6*, *Nfatc1*, *Acp5*, *Ctsk*, and *Mmp9* in the long bone of 4get and IL-5tg/4get mice ($n = 9$, 7/8). Data are shown as mean ± SEM. Symbols represent individual mice. *P* values were determined by two-tailed Mann–Whitney test (2a, 2c BV/TV, 2i, 2l Tnfrsf11a, Mmp9) or unpaired two-tailed *t* test (2c Tb.N, Tb.Th, Tb.Sp, 2e, 2f, 2g, 2k, 2l Traf6) for single comparisons. Asterisks mark statistically significant difference (*$P < 0.05$, **$P < 0.01$, ***$P < 0.001$, ****$P < 0.0001$). Source data are provided as a Source data file.

on bone homeostasis. Therefore, we proceeded to examine the bone characteristics of eosinophil-deficient ΔdblGATA mice. The GATA-1 transcription factor is essential for the differentiation of various hematopoietic lineages, including eosinophils, mast cells, erythrocytes, and platelets[22]. The ΔdblGATA mice used in this study have a selective loss of eosinophils due to the deletion of the high-affinity GATA-binding site in the GATA-1 promoter[23]. As anticipated, the bone marrow of ΔdblGATA mice showed an absence of the eosinophil population (Siglec-F⁺CD11b⁺Ly6GᶦⁿᵗCD45⁺ cell) and only modest alterations in the basophil, erythroid and megakaryocyte compartment, while all other GATA-1-expressing lineages were unaffected (Fig. 2a and Sup. Figures 1 and 2). Analysis through micro-computed

tomography (μCT) and histomorphometry revealed a significant reduction in bone mass in eosinophil-deficient mutants compared to wild-type (WT) controls (Fig. 2b, c). The osteoporotic bone phenotype observed in the absence of eosinophils was accompanied by a notable increase in osteoclast surface and number in TRAP-stained tibial sections (Fig. 2d, e) as well as elevated mRNA levels of key osteoclast-related genes, such as *Nfatc1* and *Acp5* (encoding for TRAP) in long bones (Fig. 2f). The osteoporotic phenotype resulting from eosinophil loss was also evident at older age (5 months) as shown by reduced bone volume and increased mRNA expression of *Tnfrsf11a* (encoding for RANK) and *Ctsk* in the mutant mice compared to WT controls (Sup. Figure 3a–c). Notably, an increase in bone volume and a substantial

reduction in the number of osteoclasts were observed in ΔdblGATA mice two weeks after reconstitution with $3 \times 10^6$ eosinophils by intravenous injection (Sup. Figure 3d–g).

To extend these observations, we took advantage of IL-5tg/4get (IL-5 transgenic, IL-4/GFP-enhanced transcript (4get)) mice. Even though the IL-5 receptor (IL-5Rα/CD125) is not exclusively expressed on eosinophils and is also present on basophils, B1 cells and neutrophils[24], we demonstrated that these mice predominantly display a hyper-eosinophilic phenotype (Fig. 2g and Sup. Figures 1 and 4). Of note, IL-5tg/4get mice showed significantly increased steady-state bone mass compared to 4get littermate controls (Fig. 2h, i). In accordance, this phenotype was associated with reduced osteoclast surface and number in TRAP-stained tibial sections (Fig. 2j, k) as well as impaired mRNA expression of osteoclastogenic genes, like *Tnfrsf11a*, *Traf6*, and *Mmp9* in bone tissue (Fig. 2l). Moreover, we investigated whether IL-5 could potentially affect bone homeostasis independently of eosinophils. Thus, we treated eosinophil-deficient ΔdblGATA mice with IL-5 to investigate whether this intervention could rescue their bone phenotype. However, the intraperitoneal injection of 500 ng of recombinant murine IL-5 daily for two consecutive weeks had no discernible effect on the bone and osteoclast phenotype of ΔdblGATA mice (Sup. Figure 3h–k). Altogether, these results support the concept that eosinophils participate in bone homeostasis by negatively regulating bone-resorbing osteoclasts.

## Eosinophils impair osteoclast differentiation and function in vitro

Our in vivo analyses suggest that eosinophils inhibit osteoclast development. First, we excluded that bone marrow derived monocytes (BMMs) from eosinophil-deficient ΔdblGATA mice have a deficit to differentiate into osteoclasts. When differentiating BMMs with 20 ng/mL M-CSF and 10 ng/mL RANKL into osteoclasts in vitro, no differences could be observed between WT and ΔdblGATA osteoclasts regarding the number of differentiated cells and the expression of osteoclastogenic genes (Sup. Figure 5), suggesting that the monocyte lineage is not intrinsically affected by the lack of eosinophils in the bone marrow. Moreover, these results indicate that osteoclast formation is controlled by the presence of eosinophils in the bone marrow.

Thereupon, we examined whether eosinophils interact with osteoclast precursors (OCPs) and thereby impair their differentiation into mature osteoclasts. To do so, we sorted eosinophils from hyper-eosinophilic IL-5tg/4get mice (Sup. Figure 6a, b) and co-cultured them with BMMs under osteoclastogenic conditions in vitro. BMMs were cocultured with different ratios of eosinophils on day 0 of differentiation for 48 h (Fig. 3a, b). Sorted eosinophils still presented a viability of up to 80% after the culture for 48 h (Sup. Figure 6c). TRAP staining was performed on day 3 of differentiation and osteoclasts with equal or more than 3 and equal or more than 5 nuclei were counted (Fig. 3a, b). A dose dependent decrease of osteoclasts was detected for both moderate- and high-nucleated osteoclasts with increasing numbers of eosinophils, while the viability of BMMs was unaffected (Fig. 3a, b). Moreover, osteoclast-associated genes were strongly downregulated following the stimulation with eosinophils (Sup. Figure 6d). In contrast, neutrophils, another granulocyte population, did not exhibit any discernible influence on the number of osteoclasts or the expression of osteoclast-relevant genes (Sup. Figure 7a–e). Interestingly, we observed that eosinophils displayed an anti-osteoclastogenic function only when co-cultured with osteoclast precursors and not when co-cultured with more mature osteoclasts (Sup. Figure 6f–h).

A dose dependent reduction of osteoclasts (with ≥3 and ≥5 nuclei) and a decrease in osteoclast-related gene expression was also observed after treating BMMs with increasing amounts of eosinophil supernatant (Eos spn) without affecting the viability of the cells (Fig. 3c, d and Sup. Figure 6e), while the addition of control

supernatant collected from BMMs had no impact on the differentiation of osteoclasts or the expression of osteoclast-associated genes (Sup. Figure 7f–h). Notably, even when eosinophils and BMMs were physically separated by a membrane with a pore size of 1 μm during the co-culture, eosinophils inhibited the formation of osteoclasts (Sup. Figure 6i, j). A reduction in osteoclasts and a significantly decreased expression of the osteoclast-related gene *Acp5* were also detectable when more mature osteoclasts were supplemented with eosinophil supernatant (Sup. Figure 6f–h). These observations suggest that the inhibitory effect is likely mediated by a soluble factor released by eosinophils.

To understand the molecular changes occurring in osteoclasts upon contact with eosinophils, we performed bulk RNA-sequencing of osteoclasts after stimulation with eosinophils (2:1 Eos/BMMs ratio; OCsEo) or eosinophil supernatant (1:2 dilution; OCsEoS) compared with unstimulated osteoclasts (OCsCtr). Principal component analysis (PCA) depicted the variation among the three groups, showing that the gene expression pattern changes substantially upon stimulation with eosinophils (Fig. 3e). Interestingly, osteoclasts stimulated with eosinophils or eosinophil supernatant clustered together, indicating that they have similar expression profiles (Fig. 3e). Analysis of differentially expressed genes (DEG) revealed that 484 and 519 genes were upregulated and 401 and 593 genes were downregulated in osteoclasts exposed to eosinophils and eosinophil supernatant, respectively (Fig. 3f). Interestingly, only 78 genes were differentially expressed between osteoclasts exposed to either eosinophils or eosinophil supernatant suggesting similar transcriptional changes (Fig. 3f). DEG analysis demonstrated a strong downregulation of genes important for osteoclastogenesis upon treatment with eosinophils and eosinophil supernatant, including *Csf1r*, *Oscar*, *Traf6*, *Fosl2*, and *Ocstamp* (Fig. 3g). In addition, eosinophil- and eosinophil supernatant-exposed osteoclasts showed reduced expression of oxidative phosphorylation-related genes, genes associated with cytoskeleton rearrangement, like *Itgb3*, and genes relevant for bone resorption, such as *Acp5*, *Atp6v0d2*, *Ctsk*, and *Mmp9* (Fig. 3g). In contrast, genes that are important for phagocytosis, like *Cd14*, *Cd36*, and *Trem2* were highly upregulated in osteoclasts upon contact with eosinophils or eosinophil supernatant (Fig. 3h). Interestingly, we could also detect the expression of eosinophil-related genes (*Alox15*, *Alox5ap*, *Ear1*, *Il5*, *Mpo*, *Retnlg*, *Siglecf*) in osteoclasts, when cells were cultured with eosinophils (Fig. 3h), indicating that osteoclasts might internalize eosinophils. To ensure that the eosinophil-associated genes do not distort the grouping of samples, we additionally excluded these genes from the bulk RNA-sequencing analysis (Sup. Figure 8). The PCA still indicated clear differences between the control group and osteoclasts stimulated with eosinophils or eosinophil supernatant. However, the distinction between the two stimulated groups became more equalized.

Since the DEG analysis unveiled a decreased expression of a wide range of genes associated with cytoskeleton rearrangement and bone degradation in osteoclasts upon treatment with eosinophils and eosinophil supernatant, we analyzed the demineralization activity of osteoclasts following stimulation. Eosinophil- and eosinophil supernatant-exposed osteoclasts exhibited reduced capacity to demineralize hydroxyapatite in comparison with controls (Fig. 3i, j). These in vitro experiments demonstrate that eosinophils negatively regulate osteoclast differentiation and activity probably by the secretion of specific mediators.

## Eosinophils secrete eosinophil peroxidase and reduce mitochondrial ROS level and MAP kinase activation in osteoclast precursors

Eosinophils generate and store their mediators in specific granules. The secretion of these molecules can be selectively regulated and is central for eosinophil function. Since not only the presence of eosinophils but also their supernatant was able to inhibit osteoclast

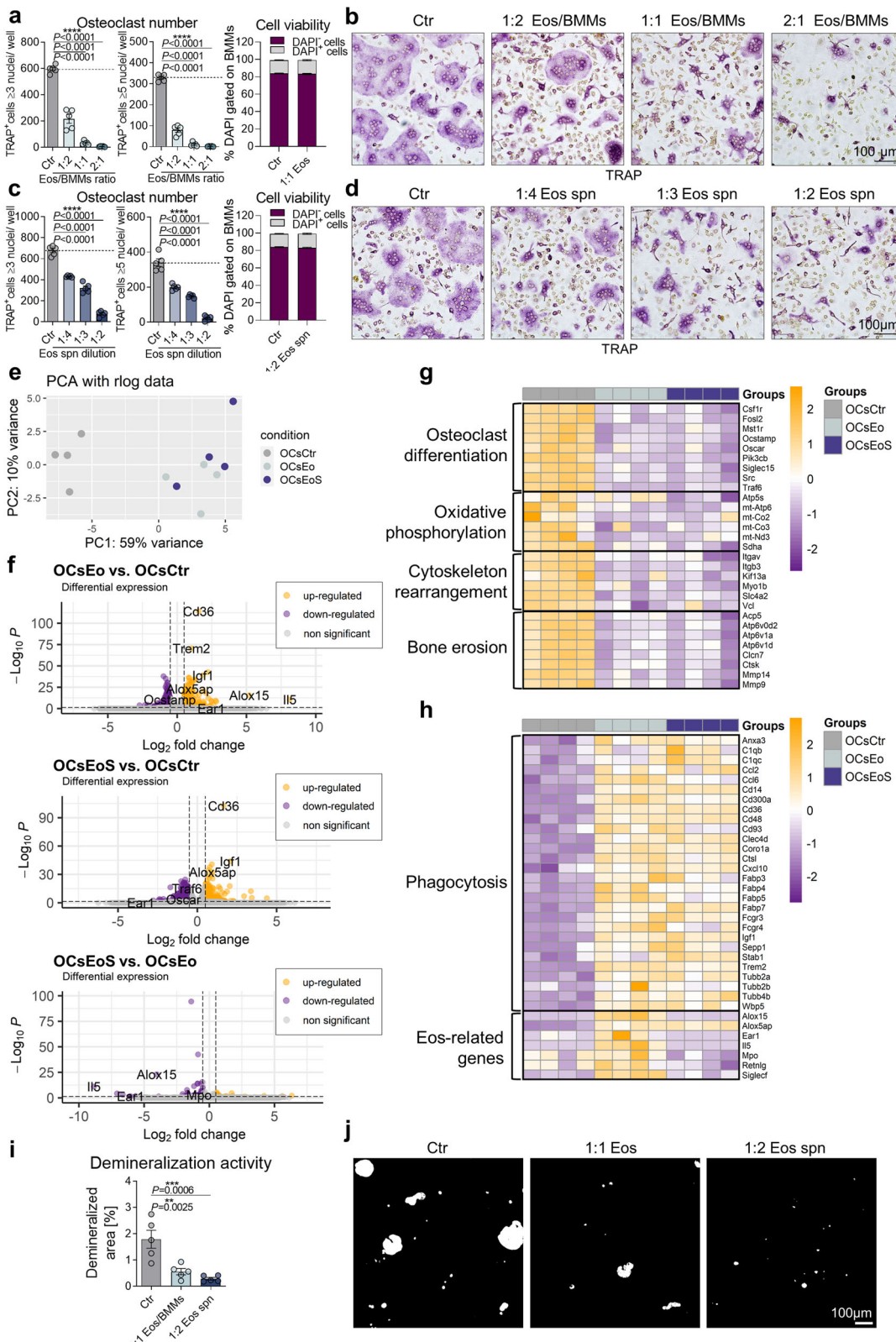

differentiation, it is likely that eosinophils release soluble mediators that facilitate this suppressive effect. Eosinophils are able to secrete IL-4 and IL-13, cytokines known for their osteoclast-inhibiting properties[25–28]. Thus, we were wondering whether the osteoclast-suppressing function of eosinophils is mediated by these cytokines. To test this hypothesis, co-culture experiments were performed with BMMs isolated from IL-4Rα and STAT6 knockout (KO) mice. These

BMMs cannot respond to IL-4 and IL-13. BMMs were stimulated with eosinophils, eosinophil supernatant or IL-4 as control. While the osteoclast-inhibitory action of IL-4 was completely abolished in BMMs isolated from IL-4Rα and STAT6 KO mice, eosinophils as well as their supernatant still inhibited osteoclast differentiation (Sup. Figure 9a, b), showing that the osteoclast-suppressing effect of eosinophils is IL-4 and IL-13 independent.

**Fig. 3 | Eosinophils inhibit osteoclast differentiation and function in vitro.**
**a, b** Quantification, viability (DAPI gated on BMMs) (**a**), and representative images
(**b**) of TRAP-positive polynucleated (≥3 nuclei or ≥5 nuclei) WT osteoclasts co-
cultured with different ratios of eosinophils (Eos) compared with unstimulated
control (*n* = 5). Scale bar, 100 μm. **c, d** Quantification, viability (DAPI gated on
BMMs) (**c**), and representative images (**d**) of TRAP-positive polynucleated (≥3
nuclei or ≥5 nuclei) WT osteoclasts co-cultured with different dilutions of eosi-
nophil supernatant (Eos spn) compared with unstimulated control (*n* = 5). Scale
bar, 100 μm. **e–h** Bulk RNA-seq. analysis of WT osteoclasts stimulated with eosi-
nophils (2:1 Eos/BMMs ratio; OCsEo) or eosinophil supernatant (1:2 dilution;
OCsEoS) compared with unstimulated control (OCsCtr) (*n* = 4). Principal compo-
nent analysis (PCA) visualizing the patterns of the individual samples of the 3
different groups (**e**). Volcano plots showing differentially expressed genes (DEG)
between the groups OCsEo vs. OCsCtr, OCsEoS vs. OCsCtr and OCsEoS vs. OCsEo
(**f**). Statistically significantly upregulated genes are orange and downregulated

genes are purple. Heat map showing osteoclast-associated DEGs between the
aforementioned groups (**g**). Heat map showing phagocytosis- and eosinophil-
associated DEGs between the aforementioned groups (**h**). **i, j** Quantification (**i**) and
representative images (**j**) of the demineralization activity of WT osteoclasts cul-
tured on hydroxyapatite-coated plates following the stimulation with eosinophils
(1:1 Eos/BMMs ratio) or eosinophil supernatant (1:2 dilution) compared with
unstimulated control (*n* = 5). Scale bar, 100 μm. Data are shown as mean ± SEM.
Symbols represent individual mice. *P* values were determined by one-way ANOVA
Dunnett's test (3a, 3c, 3i) for multiple comparisons. Asterisks mark statistically
significant difference (**P* < 0.01, ***P* < 0.001, ****P* < 0.0001). Differential expres-
sion analysis between 2 groups was performed with log2 fold change cut off 0.5.
The resulting *P* values were adjusted using the Benjamini−Hochberg approach for
controlling the false discovery rate (FDR). Genes with an adjusted *P* value ($P_{adj}$) less
than 0.05 were assigned as differentially expressed (2f−h). Source data are pro-
vided as a Source data file.

To find other possible candidates, a proteome profiler array was
carried out, where the expression of 111 different cytokines was
investigated in the supernatant of sorted eosinophils after culture for
48 h (Fig. 4a, b). Among the highly secreted mediators (Fig. 4a, b),
myeloperoxidase (MPO) and the cysteine proteinase inhibitor cystatin
C (CysC) have been previously described to downregulate osteoclast
differentiation[29,30]. Since the heme peroxidase EPX is specifically
expressed by eosinophils, is functionally and structurally very similar
to MPO and has also shown to inhibit osteoclasts[31], the amount of EPX
compared with CysC and MPO was measured in the supernatant of
eosinophils by ELISA (Fig. 4c). Indeed, eosinophils secrete the anti-
osteoclastogenic cytokines CysC, EPX, and to a lesser extent MPO.
Furthermore, eosinophils have the capability to release TGF-β[32], an
inhibitor of osteoclasts[33], which was not included in the proteome
profiler array. Consequently, we measured the secretion level of TGF-β
by sorted eosinophils and compared it to the level produced by BMMs
after culture for 48 h. The level of TGF-β produced by unstimulated
eosinophils was found to be even lower than the one from BMMs (Sup.
Figure 9c). However, to investigate the potential involvement of TGF-β
in the inhibitory effects of eosinophil supernatant, we introduced a
TGF-β signaling inhibitor during the stimulation of BMMs with eosi-
nophils and eosinophil supernatant. Inhibiting the interaction of TGF-β
with BMMs did not alter the ability of eosinophils to suppress osteo-
clast differentiation and the expression of osteoclast-related genes
(Sup. Figure 9d−f).

To investigate the relevance of the upregulated cytokines CysC,
MPO and EPX for the osteoclast-suppressing action of eosinophils,
their impact on osteoclastogenesis was compared with the effects of
eosinophil supernatant (Fig. 4d−f). While CysC had no effect on
osteoclast formation, the two heme peroxidases MPO and EPX both
reduced osteoclastogenesis in a dose dependent manner without
affecting the viability of the cells (Fig. 4d−f). Even low concentrations
of EPX were able to completely block the differentiation of osteoclasts
(Fig. 4d−f). Finally, to examine whether the osteoclast-suppressing
function of eosinophil supernatant is dependent on heme peroxidases,
we used the MPO/EPX inhibitor 4-ABAH[34] during the stimulation
experiments (Fig. 4g, h). Notably, the neutralization approaches par-
tially abolished the osteoclast-suppressing effect of eosinophil super-
natant (Fig. 4g, h), assuming that eosinophils control osteoclast
differentiation through the release of heme peroxidases.

In previous publications it has been shown that heme peroxidases
can be directly internalized by osteoclasts, thereby inhibiting RANKL-
mediated downstream signaling[31]. Recently, it was found that MPO
hampers osteoclast differentiation through intrinsic reduction of
reactive oxygen species (ROS) in osteoclast precursors[29]. Our RNA-seq.
data revealed that eosinophils and eosinophil supernatant induced the
expression of genes relevant for phagocytosis, while the expression of
ROS-related genes and especially RANK-regulated genes were inhib-
ited (Fig. 3g, h). Based on these findings, we assumed that eosinophils

suppress osteoclast differentiation and function through heme
peroxidase-mediated ROS reduction and subsequent inhibition of
RANK signaling. Of note, flow cytometry revealed that mitochondrial
(mt) ROS expression was significantly reduced in osteoclast precursors
upon contact to eosinophils or eosinophil supernatant (Fig. 4i, j). EPX
was able to decrease mtROS in a dose dependent manner, while MPO
had no significant impact (Fig. 4i, j), suggesting that EPX is the major
anti-osteoclastogenic heme peroxidase. Further on, we investigated
the RANK signaling in osteoclasts exposed to eosinophils, eosinophil
supernatant and EPX. Therefore, lysates of osteoclasts were generated
upon cultivation with the different stimuli and assessed for phos-
phorylation of MAP kinases downstream of RANK activation (Fig. 4k, l).
These analyses revealed a reduced level of phospho-JNK and phospho-
p38 in osteoclasts following stimulation with eosinophils, while cells
exposed to eosinophil supernatant and EPX showed a tendential
decrease (Fig. 4k, l). Thus, mechanistically, eosinophils reduce ROS
concentration in osteoclast precursors, thereby inhibiting RANK-
mediated MAP kinase activation and osteoclast differentiation.

## Loss of eosinophils amplifies postmenopausal and inflammatory bone loss

As we could show that eosinophils possess osteoclast-regulating
properties in steady-state conditions, we wanted to examine their role
during pathological bone loss. Hence, we investigated the impact of
eosinophils on two different disease models, hormone-induced bone
loss by ovariectomy (OVX) and inflammatory bone loss after K/BxN
serum transfer arthritis (STA). As the axial bone loss is not well docu-
mented for the STA model, we directed our focus towards assessing
bone decline specifically in the appendicular skeleton.

First, OVX was performed in eosinophil-deficient mice (Fig. 5a, b
and Sup. Fig. 10a). μCT and histological analysis revealed an amplified
bone decline upon OVX in eosinophil-deficient mice as compared with
WT controls (Fig. 5c, d). Enhanced bone loss in OVX ΔdblGATA mice
was associated with increased osteoclasts in long bones, while the
number of osteoclast precursors in the bone marrow was not altered
(Fig. 5b, e, f and Sup. Fig. 11). In line with our in vitro data, eosinophil-
deficient mice showed a strongly decreased mRNA expression of *Epx*
in long bones in the sham and the OVX group (Fig. 5g). Thus, the lack of
eosinophils results in an enhanced hormone-induced bone loss.

As a next step, we conducted STA experiments using mice with
eosinophil deficiency (Fig. 5h, i). As previously published[35], eosinophil-
deficient mice showed a more severe disease progression (Sup.
Fig. 10b). In addition, eosinophil-deficient mice presented enhanced
inflammatory bone loss with elevated numbers of bone-resorbing
osteoclasts (Fig. 5j−m). In this case, also the number of osteoclast
precursors was strongly increased in the mutant mice compared with
controls upon STA (Fig. 5i). In accordance, the *Epx* mRNA expression
was completely suppressed in long bones of ΔdblGATA mice com-
pared with control mice with and without STA (Fig. 5n). As a

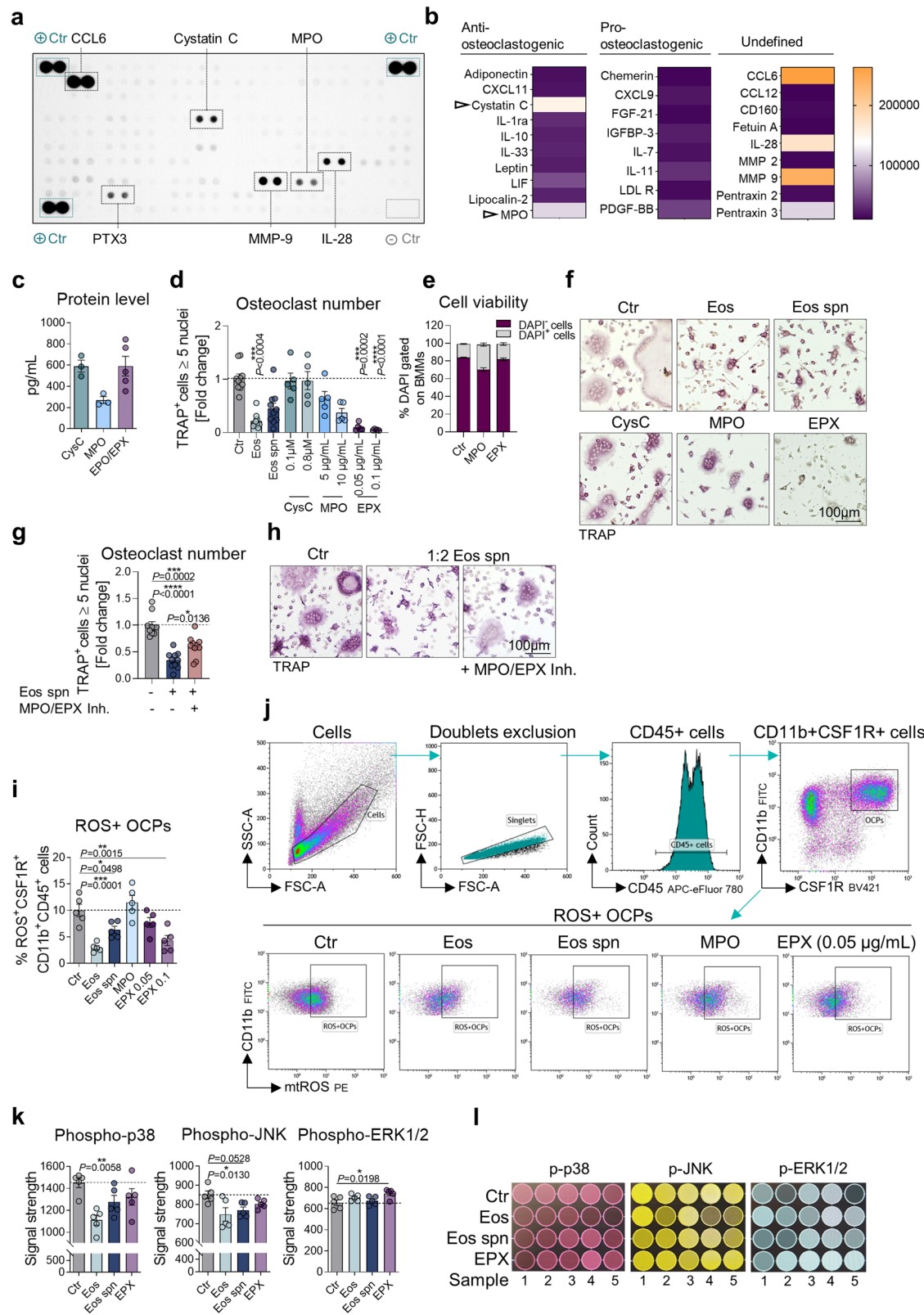

conclusion, loss of eosinophils is also associated with a more severe inflammatory bone loss.

### High eosinophil numbers reduce postmenopausal and inflammatory bone loss

We next investigated hormone- and inflammation-induced bone loss in hyper-eosinophilic IL-5tg/4get mice. These mice were completely protected from estrogen deficiency-mediated bone loss as the bone mass as well as the number of osteoclasts was not affected by the removal of the ovaries in comparison with littermate 4get controls (Fig. 6a–f and Sup. Fig. 10c). The number of osteoclast precursors in the bone marrow was decreased in IL-5tg/4get mice independent of OVX surgery (Fig. 6b and Sup. Fig. 11). In line with these findings, IL-5tg/4get mice presented a highly

**Fig. 4 | Eosinophil-dependent osteoclast supression is mediated by eosinophil peroxidase, reducing mitochondrial ROS levels and inhibiting RANKL-mediated signaling in osteoclast precursors. a, b** Proteome profiler array with supernatant from sorted eosinophils collected after 48 h culture (**a**) and heat map of relative levels (integrated pixel density) of the detected cytokines grouped into anti-osteoclastogenic, pro-osteoclastogenic, and undefined role in osteoclastogenesis (**b**). **c** Protein level of cystatin C (CysC), myeloperoxidase (MPO), and eosinophil peroxidase (EPO/EPX) in eosinophil supernatant ($n = 3, 3, 5$). **d–f** Quantification (**d**), viability (DAPI gated on BMMs) (**e**), and representative images (**f**) of TRAP-positive polynucleated (≥5 nuclei) WT osteoclasts supplemented with eosinophils (1:1 Eos/BMMs ratio), eosinophil supernatant (1:2 dilution), 0.1–0.8 μM of CysC, 5–10 μg/mL of MPO, and 0.05–0.1 μg/mL of EPX compared with unstimulated control ($n = 10$, 10, 10, 5, 5, 5, 5, 5). Scale bar, 100 μm. **g, h** Quantification (**g**) and representative images (**h**) of TRAP-positive polynucleated (≥5 nuclei) WT osteoclasts supplemented with eosinophil supernatant (1:2 dilution) in the presence or absence of 10 μM of MPO/EPX inhibitor compared with unstimulated control ($n = 10$). Scale bar, 100 μm. **i, j** Quantification (**i**) and gating strategy (**j**) of mitochondrial (mt) ROS-positive osteoclast precursors (mtROS$^+$CSF1R$^+$CD11b$^+$CD45$^+$ cells) analyzed by FC at day 2 of differentiation after stimulation with eosinophils (1:1 Eos/BMMs ratio), eosinophil supernatant (1:2 dilution), 10 μg/mL of MPO, and 0.05–0.1 μg/mL of EPX compared with unstimulated control ($n = 5$). **k, l** Quantification (**k**) and representative signal (**l**) of phosphorylated p38 (Thr180/Tyr182), phosphorylated JNK (Thr183/Tyr185), and phosphorylated ERK1/2 (Thr202/Tyr204; Thr185/Tyr187) in whole cell lysates of osteoclasts supplemented with eosinophils (1:1 Eos/BMMs ratio), eosinophil supernatant (1:2 dilution), or 1 μg/mL of EPX compared with unstimulated control ($n = 5$). Data are shown as mean ± SEM. Symbols represent individual mice. $P$ values were determined by Kruskal–Wallis Dunn's test (4d, 4k phospho-p38), one-way ANOVA Tukey's test (4g) or one-way ANOVA Dunnett's test (4i, 4k phospho-JNK, phospho-ERK1/2) for multiple comparisons. Asterisks mark statistically significant difference (*$P < 0.05$, **$P < 0.01$, ***$P < 0.001$, ****$P < 0.0001$). Source data are provided as a Source data file.

elevated mRNA level of *Epx* in long bones compared with 4get controls (Fig. 6g).

Similarly, IL-5tg/4get mice showed reduced arthritis severity (Sup. Fig. 10d), as previously shown[35], but were also completely protected from inflammatory bone loss (Fig. 6h–m). While 4get control mice presented reduced bone mass and high numbers of osteoclast precursors and mature osteoclasts in long bones after STA, IL-5tg/4get mice were unaffected by the arthritis-induced bone loss (Fig. 6i–m). In accordance, the mRNA expression of *Epx* in bone tissue was strongly upregulated in IL-5tg/4get mice especially after STA as compared with littermate 4get controls (Fig. 6n). Summarizing, a high eosinophil number is able to protect from pathological osteoclast-mediated bone loss.

### EPX treatment ameliorates inflammation-mediated bone loss

Our in vitro experiments and in vivo disease models indicate that EPX might be the key driver in eosinophil-dependent osteoclast inhibition and reduction of bone loss. To investigate the beneficial role of EPX during inflammation-mediated bone loss, BALB/c WT mice were treated with EPX at day 0 and 4 of K/BxN serum transfer arthritis (Fig. 7a). EPX treatment led to an enhanced resolution of arthritis (Sup. Fig. 10e). Notably, EPX-treated mice displayed a less severe progression of bone loss (Fig. 7c, d), while the number of eosinophils and osteoclast precursors was not significantly altered (Fig. 7b). This bone phenotype can be explained by a significantly reduced surface and number of mature osteoclasts in long bones of EPX-treated mice with arthritis compared to vehicle-treated mice with arthritis (Fig. 7e, f). Thus, EPX represents a potent anti-osteoclastogenic enzyme, preventing excessive bone resorption by osteoclasts.

### Eosinophils are associated with bone mass in humans

As a translational approach, we determined the impact of eosinophils on bone loss in humans by analyzing rheumatoid arthritis (RA) patients as osteoporosis is one of the most frequent comorbidities of RA[36]. Immunofluorescence staining of RA synovial tissue confirmed the localization of CD68-positive, multinucleated osteoclasts with EPX-positive eosinophils in proximity to mineralized bone (Fig. 8a). Besides, we investigated the mRNA expression of the eosinophil-specific gene *RNASE2* in the blood of 34 RA patients. Since the monocytic precursors of osteoclasts enter the bone through the bloodstream and commonly express genes associated with osteoclasts[37], we analyzed the correlation between *RNASE2* expression and the osteoclast genes *NFATC1* and *ACP5* (Fig. 8b). While no correlation could be detected regarding the expression of *NFATC1*, the expression of *ACP5* was negatively correlated with the expression of eosinophil-specific *RNASE2* (Fig. 8b). Finally, we analyzed the relation of circulating eosinophil numbers and serum levels of eosinophil cationic protein (ECP) to bone mass in RA patients and healthy

controls. We analyzed human bone by high-resolution peripheral quantitative computed tomography (HR-pQCT), which allows separately measuring trabecular and cortical bone with high accuracy. In order to ensure a homogeneous cohort and minimize the impact of treatment variables, we focused on early RA patients receiving methotrexate monotherapy with no allowance for other medications. We further divided participants into two groups: one without any supplementary glucocorticoid treatment and another receiving glucocorticoids considering the well-established influence of glucocorticoids on eosinophils. RA patients with ($n = 25$) and without ($n = 25$) glucocorticoids and healthy controls ($n = 25$) were well matched with respect to age, sex and body mass index (Table 1). RA patients without and with glucocorticoids had an overall short disease duration (1.5 and 1.6 years) and low disease activity (DAS28 score 3.18 and 3.26 units). Trabecular and cortical bone mass were lower in RA patients than in healthy controls. Trabecular but not cortical bone mass was significantly correlated to eosinophil counts and ECP levels in RA patients without glucocorticoids ($r = 0.71$, $p = 0.0001$; $r = 0.55$, $p = 0.0039$) and in healthy controls ($r = 0.57$, $p = 0.0025$; $r = 0.57$, $p = 0.0027$). No such correlation was found in RA patients exposed to glucocorticoids ($r = 0.21$, $p = 0.31$; $r = 0.07$, $p = 0.73$) (Fig. 8c–f). Altogether, these findings postulate that eosinophils might also play a role in human bone homeostasis.

## Discussion

This work reveals that eosinophils represent important cells in bone homeostasis by inhibiting osteoclast-mediated bone resorption through EPX secretion (Fig. 9). In both mice and men, the number of eosinophils and eosinophil products are associated with higher bone mass and reduced osteoclast activity. While the steady state effect of eosinophils on bone metabolism may appear modest, it becomes evident with induced conditions such as inflammatory arthritis and sex hormone deprivation.

It is generally believed that eosinophils evolved in vertebrates across taxa as end-stage effector cells for host defense against parasites and other pathogens[38]. As anti-microbial response, eosinophils release the cytotoxic granule proteins EPX, ECP, major basic protein (MBP) and eosinophil derived neurotoxin (EDN)[39]. Dysregulated eosinophils contribute to the pathogenesis of various diseases, including asthma, eosinophilic gastrointestinal diseases and hyper-eosinophilic syndromes via the excessive secretion of granule proteins[40].

The "LIAR" (Local Immunity And/or Remodeling/Repair) hypothesis implies that the function of eosinophils is far more expansive and complex than previously appreciated, considering these cells as important regulators of tissue homeostasis[41]. As proof of this, phylogenetic studies revealed that invertebrates already exhibit "eosinophilic" hemocytes. In these species eosinophil-like cells control tissue remodeling, like metamorphosis[42]. In humans and rodents, eosinophils

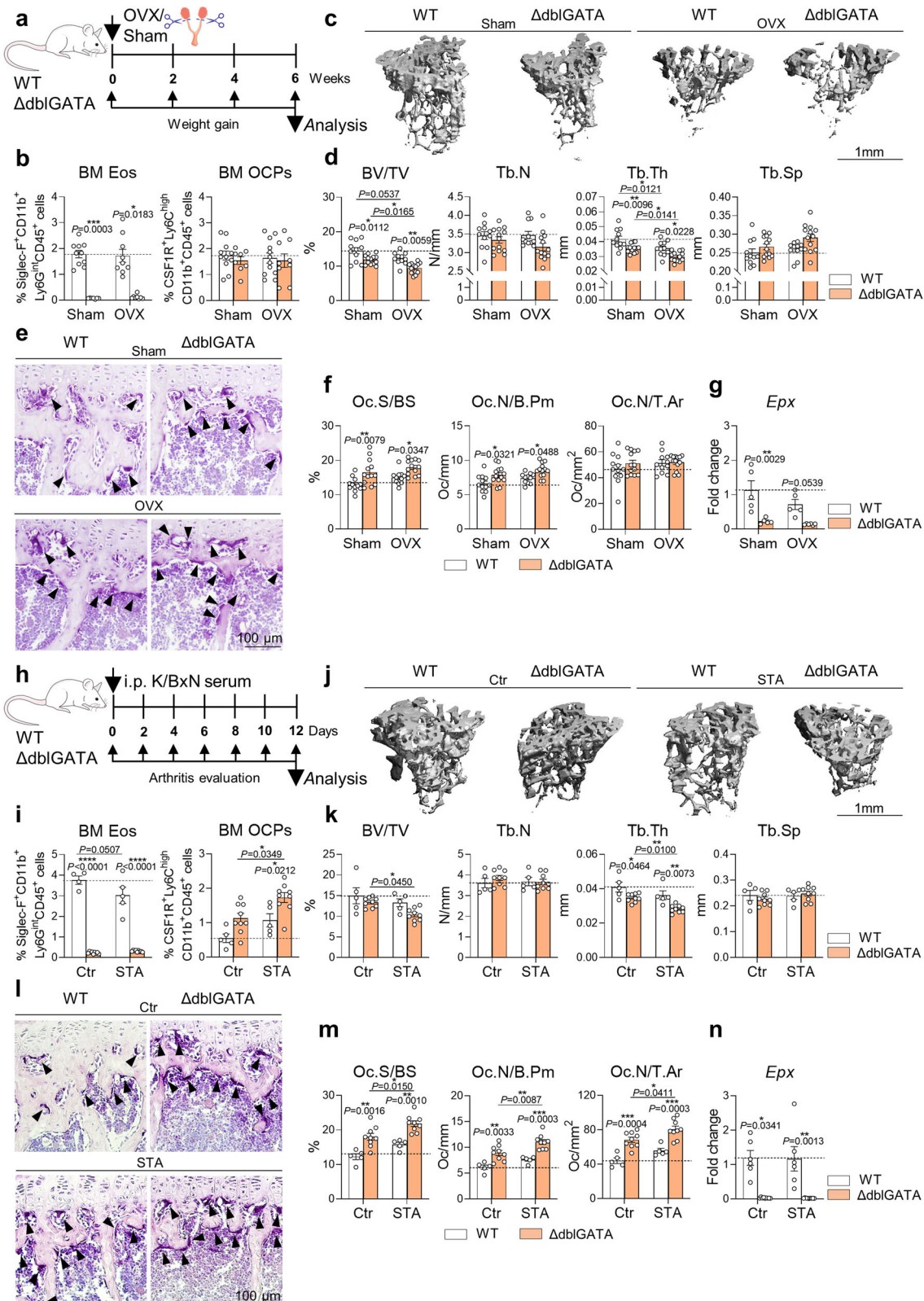

can be found in multiple organs, such as the bone marrow, the gastrointestinal tract, the lungs, the adipose tissue, the thymus, the uterus and the mammary glands, where they implement tissue homeostatic functions[43]. For instance, eosinophils have been shown to modulate macrophage polarization towards an alternative phenotype by IL-4 and IL-13, leading to the formation of brown adipose tissue[44]. Indeed, eosinophils are not necessarily pro-inflammatory as recently our own

data revealed the presence of a homeostatic eosinophil subset in joints that promotes the resolution of inflammatory arthritis. This effect was largely mediated by an eosinophil-dependent stimulation of anti-inflammatory macrophage differentiation[19,35].

In the current study, we could demonstrate that eosinophils influence the function of bone-resorbing osteoclasts. Bulk RNA-seq. data confirmed the downregulation of genes essential for osteoclast

**Fig. 5 | Eosinophil loss exaggerates bone loss in response to sex hormone deprivation and inflammation. a** Experimental outline of ovariectomy (OVX) in WT vs. ΔdblGATA mice. **b** Percentage of eosinophils (Siglec-F⁺CD11b⁺Ly6G^int CD45⁺ cells) (*n* = 10, 9, 7, 8) and osteoclast precursors (OCPs/CSF1R⁺Ly6C^high CD11b⁺CD45⁺ cells) (*n* = 11, 12, 8, 9) in the BM of WT sham and OVX mice vs. ΔdblGATA sham and OVX mice. **c, d** 3D trabecular structure by μCT (**c**) and quantification of bone volume per total volume (BV/TV), trabecular number (Tb.N), trabecular thickness (Tb.Th), and trabecular separation (Tb.Sp) (**d**) in tibial bone from the aforementioned 4 groups (*n* = 11, 11, 13, 13). Scale bar, 1 mm. **e, f** Representative TRAP staining (**e**) and quantification of osteoclast surface per bone surface (Oc.S/BS), osteoclast number per bone perimeter (Oc.N/B.Pm), and osteoclast number per tissue area (Oc.N/T.Ar) (**f**) in tibial bone sections from the aforementioned 4 groups (*n* = 12, 11, 13, 13). Triangles illustrate osteoclasts. Scale bar, 100 μm. **g** mRNA expression of *Epx* in long bone in the aforementioned 4 groups (*n* = 5, 5, 5, 6). **h** Experimental outline of serum

transfer arthritis (STA) in WT vs. ΔdblGATA mice. **i** Percentage of eosinophils (*n* = 4, 5, 9, 9) and OCPs (*n* = 5, 5, 9, 9) in the BM of control and STA-induced WT mice vs. control and STA-induced ΔdblGATA mice. **j, k** μCT 3D trabecular structure (**j**) and quantification of BV/TV, Tb.N, Tb.Th, and Tb.Sp (**k**) in tibial bone from the aforementioned 4 groups (*n* = 5, 5, 9, 9). Scale bar, 1 mm. **l, m** Representative TRAP staining (**l**) and quantification of Oc.S/BS, Oc.N/B.Pm, and Oc.N/T.Ar (**m**) in tibial bone sections from the aforementioned 4 groups (*n* = 5, 5, 9, 9). Triangles illustrate osteoclasts. Scale bar, 100 μm. **n** mRNA expression of *Epx* in long bone in the aforementioned 4 groups (*n* = 6, 6, 9, 9). Data are shown as mean ± SEM. Symbols represent individual mice. *P* values were determined by Kruskal–Wallis Dunn's test (5b, 5n) or two-way ANOVA Tukey's test (5d, 5f, 5g, 5i, 5k, 5m) for multiple comparisons. Asterisks mark statistically significant difference (*\**P* < 0.05, \*\**P* < 0.01, \*\*\**P* < 0.001, \*\*\*\**P* < 0.0001). Source data are provided as a Source data file.

signaling, while phagocytosis pathways were up-regulated after contact of osteoclasts with eosinophils or eosinophil supernatant. The osteoclast-suppressing action of eosinophils was not mediated by IL-4, IL-13 or TGF-β, but rather by the secreted heme peroxidase EPX. Neutrophil heme peroxidase MPO[45] and eosinophil heme peroxidase EPX[39] originally catalyze halides and $H_2O_2$ to water and oxidized substrates. These oxidized substrates, also known as ROS, kill invading pathogens[46,47]. However, several studies have shown that heme peroxidases also have immune regulatory properties[34,48,49]. A few studies also addressed their role in osteoclasts. For instance, Zhao et al. demonstrated that MPO, which is structurally and functionally very similar to EPX, impairs osteoclast development by decreasing the intracellular level of $H_2O_2$ and ROS[29]. ROS is an important inducer of osteoclast differentiation and function, activating the MAPK and NF-κB signaling pathways[50,51]. Consequently, MPO deficient mice presented decreased bone mass and increased osteoclast numbers in steady-state conditions[29]. In accordance, we found that EPX also decreases the level of mitochondrial ROS in osteoclast precursors and that eosinophil-deficient ΔdblGATA mice lacking relevant EPX production also develop bone loss based on increased osteoclast activity. Interestingly, when the GATA-1 transcription factor is specifically depleted in megakaryocytes (MKs), leading to mutant MKs, an increased bone volume was reported due to the activation of osteoblasts by the mutant MKs[52].

Further studies have shown that heme peroxidases can also be internalized by osteoclast precursors, where they impair RANKL-mediated MAPK phosphorylation and thereby osteoclast activation[31]. In accordance, we could show that eosinophils diminished JNK and p38 MAPK phosphorylation in osteoclasts, thereby inhibiting osteoclast differentiation and function.

So far, the role of eosinophils during human bone homeostasis has been unclear. Few studies addressed the impact of eosinophilic diseases, such as asthma or eosinophilic esophagitis on bone mineral density[53–55]. However, these diseases are heavily treated with glucocorticoids, which have detrimental effects on bone, increasing the risk for osteoporosis and fractures[56]. Thus, no clear conclusion can be made from the present studies and it may be challenging to determine the impact of eosinophils on bone homeostasis in these individuals.

Osteoporosis is a frequent comorbidity of RA patients. Our study confirmed that both trabecular and cortical bone mass are lower in RA patients compared to healthy controls. We observed a significant correlation between eosinophil counts and ECP levels with trabecular bone mass in RA patients and in healthy controls. However, with glucocorticoid treatment, eosinophils and ECP levels were generally suppressed and no relation to bone mass could be detected. These observations indicate that eosinophils also play a role in the regulation of bone remodeling in humans.

Altogether, we could delineate that eosinophils possess osteoprotective functions through inhibition of bone-resorbing osteoclasts.

Secretion of EPX reduces intracellular ROS level in osteoclast precursors, impairing RANKL-mediated osteoclastogenesis. Thus, eosinophils represent a hitherto unrecognized connection between the innate immune system and bone, emphasizing the tissue homeostatic role of eosinophils in the bone compartment.

## Methods

### Patient characteristics

Peripheral blood and synovial samples (ultrasound-guided needle biopsy) were taken from RA patients that fulfil the 2010 defined RA classification criteria of the American College of Rheumatology (ACR)/ European League Against Rheumatism (EULAR)[57]. Disease activity was determined by disease activity score 28 (DAS28) based on erythrocyte sedimentation rate (ESR)[58]. RA primarily impacts females, with a female-to-male ratio of 3:1. As a result, the patient group generally comprises a higher proportion of women compared to men. Blood samples for RNA isolation were obtained from RA patients with different disease activity (21 females, 13 males; mean ± SD age: 61.7 ± 11.5 years; mean ± SD disease activity score 28: 4.09 ± 1.6). For high-resolution peripheral quantitative computed tomography (HR-pQCT) eosinophil counts, and serum levels of eosinophil cationic protein (ECP), we recruited patients with a diagnosis of RA according to the ACR/EULAR criteria[57] as well as healthy controls from the TARDA database[59]. In order to ensure a uniform participant cohort and minimize the impact of treatment variables, we selected RA patients who received methotrexate monotherapy (early RA), with no treatment with other conventional synthetic, targeted synthetic, or biologic disease-modifying anti-rheumatic drugs (DMARDs). Furthermore, patients receiving additional glucocorticoid treatment were separately analyzed from RA patients receiving no glucocorticoid treatment. Demographic characteristics (age, sex and body mass index), disease-specific characteristics (disease duration and DAS 28 score), bone parameters (trabecular and cortical bone volume), and eosinophil parameters (eosinophil counts and ECP levels) of the healthy controls and RA patients are listed in Table 1. Healthy controls were sex and age matched. Healthy controls and RA patients provided written informed consent to participate in the present study. All analyses were approved by the institutional review board (IRB) of the University Clinic Erlangen and were in accordance with the Declaration of Helsinki–Ethical Principles for Medical Research Involving Human Subjects. For the blood donation, participants received financial compensation.

### Mice

Wild-type BALB/cJRj were purchased from Janvier Labs. ΔdblGATA[23], IL-5tg/4get[60], IL-4Rα knockout[61], and STAT6 knockout[62] mice were on BALB/c background. Cx3cr1^cre R26-tdTomato mice[63] were on BL/6 background. Steady-state bone phenotype was determined at an age of 10–12 weeks including male and female mice as the BV/TV in the proximal tibia is comparable in male and female BALB/c mice in the young adult phase (2–6 month)[64]. All mice were housed in a specific

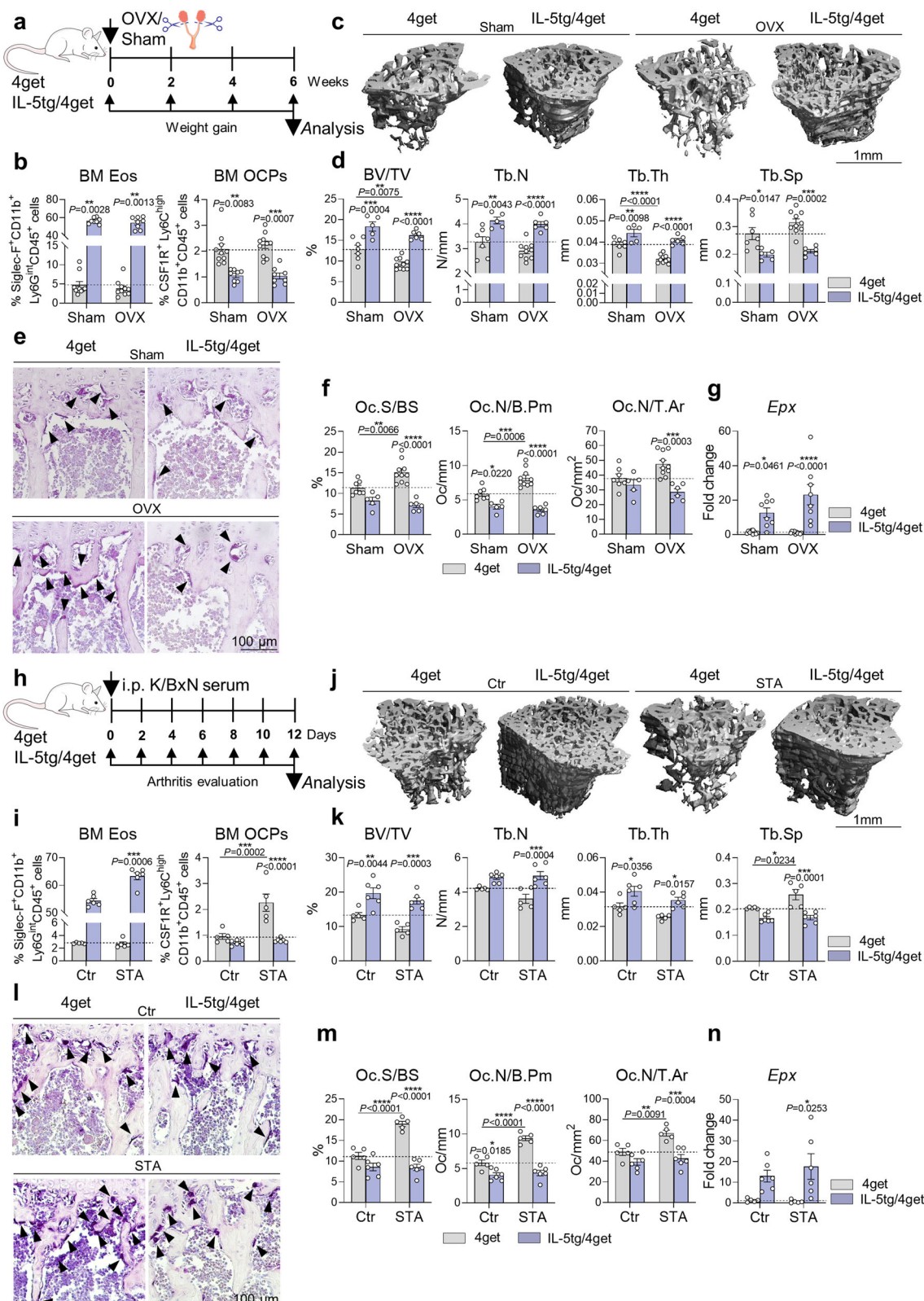

pathogen-free (SPF) facility with a temperature between 22 °C and 23 °C, humidity of 50–60%, and a regulated 12-h light/dark cycle with free access to normal diet food (ssniff, V1534-000) and water. Mice were sacrificed through increasing $CO_2$ concentration. All experiments were performed according to the rules and regulations of the animal facility FPZ (Franz-Penzoldt-Zentrum, Erlangen). Animal studies were approved by the ethics committees of the government of Unterfranken (Regierung von Unterfranken, Germany).

## Study design

The sample size used in the experiments followed the common practice of the field[65] and was verified by G*Power (version 3.1.9.7). Mice

**Fig. 6 | Hyper-eosinophilia reduces pathological bone loss. a** Experimental outline of ovariectomy (OVX) in 4get controls vs. IL-5tg/4get mice. **b** Percentage of eosinophils (Siglec-F⁺CD11b⁺Ly6G^intCD45⁺ cells) and osteoclast precursors (OCPs/CSF1R^highLy6C^highCD11b⁺CD45⁺ cells) in the BM of 4get sham and OVX mice vs. IL-5tg/4get sham and OVX mice ($n = 10, 10, 8, 8$). **c, d** 3D trabecular structure by μCT (**c**) and quantification of bone volume per total volume (BV/TV), trabecular number (Tb.N), trabecular thickness (Tb.Th), and trabecular separation (Tb.Sp) (**d**) in tibial bone from the aforementioned 4 groups ($n = 7, 10, 5, 6$). Scale bar, 1 mm. **e, f** Representative TRAP staining (**e**) and quantification of osteoclast surface per bone surface (Oc.S/BS), osteoclast number per bone perimeter (Oc.N/B.Pm), and osteoclast number per tissue area (Oc.N/T.Ar) (**f**) in tibial bone sections from the aforementioned 4 groups ($n = 7, 10, 5, 6$). Triangles illustrate osteoclasts. Scale bar, 100 μm. **g** mRNA expression of *Epx* in long bone in the aforementioned 4 groups ($n = 10, 10, 8, 8$). **h** Experimental outline of serum

transfer arthritis (STA) in 4get controls vs. IL-5tg/4get mice. **i** Percentage of eosinophils and OCPs in the BM of control and STA-induced 4get mice vs. control and STA-induced IL-5tg/4get mice ($n = 5, 5, 6, 6$). **j, k** μCT 3D trabecular structure (**j**) and quantification of BV/TV, Tb.N, Tb.Th, and Tb.Sp (**k**) in tibial bone from the aforementioned 4 groups ($n = 5, 5, 6, 6$). Scale bar, 1 mm. **l, m** Representative TRAP staining (**l**) and quantification of Oc.S/BS, Oc.N/B.Pm, and Oc.N/T.Ar (**m**) in tibial bone sections from the aforementioned 4 groups ($n = 5, 5, 6, 6$). Triangles illustrate osteoclasts. Scale bar, 100 μm. **n** mRNA expression of *Epx* in long bone in the aforementioned 4 groups ($n = 5, 5, 6, 6$). Data are shown as mean ± SEM. Symbols represent individual mice. *P* values were determined by Kruskal–Wallis Dunn's test (6b, 6i BM Eos) or two-way ANOVA Tukey's test (6d, 6f, 6g, 6i BM OCPs, 6k, 6m, 6n) for multiple comparisons. Asterisks mark statistically significant difference (*$P < 0.05$, **$P < 0.01$, ***$P < 0.001$, ****$P < 0.0001$). Source data are provided as a Source data file.

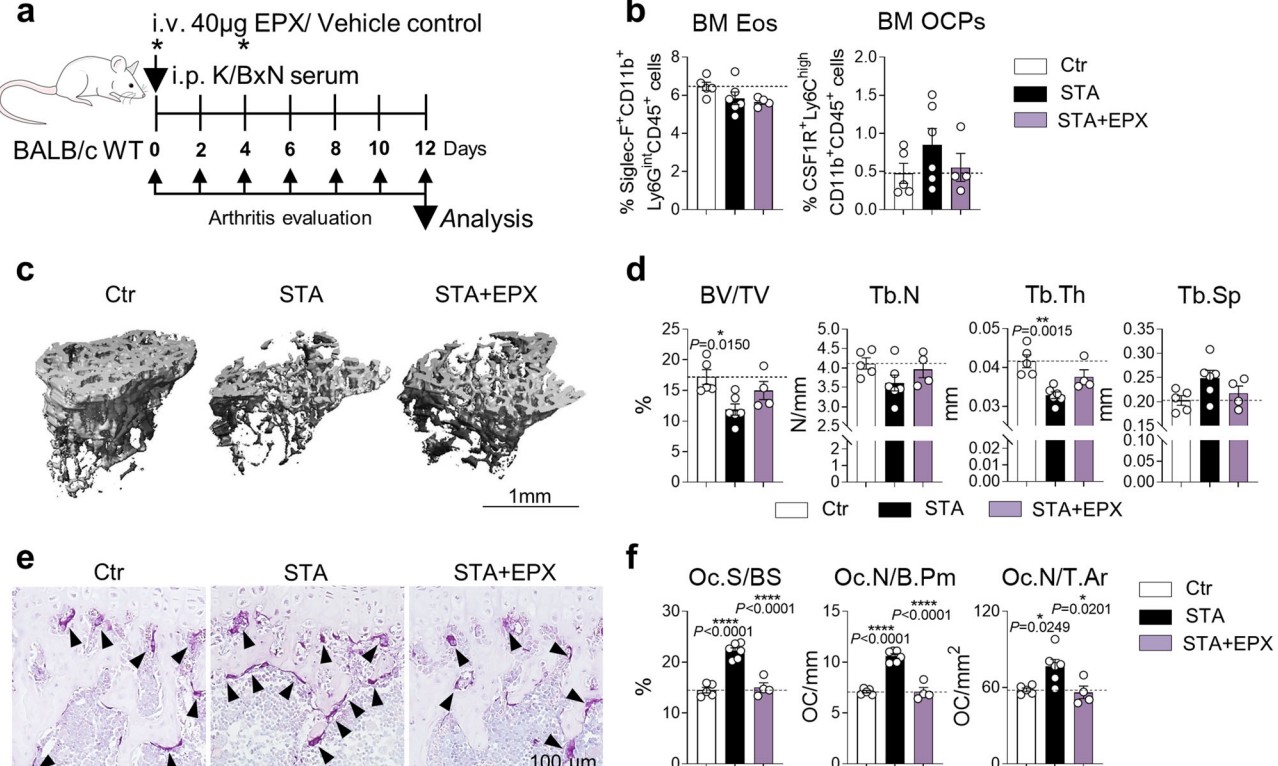

**Fig. 7 | EPX treatment ameliorates inflammation-mediated bone loss. a** Experimental outline of serum transfer arthritis (STA) in BALB/c WT mice with vehicle control or EPX treatment at day 0 and 4 post STA. **b** Percentage of eosinophils (Siglec-F⁺CD11b⁺Ly6G^intCD45⁺ cells) and osteoclast precursors (OCPs/CSF1R^highLy6C^highCD11b⁺CD45⁺ cells) in the BM of Ctr, STA, and STA + EPX mice ($n = 5, 6, 4$). **c, d** 3D trabecular structure by μCT (**c**) and quantification of bone volume per total volume (BV/TV), trabecular number (Tb.N), trabecular thickness (Tb.Th), and trabecular separation (Tb.Sp) (**d**) in tibial bone from the aforementioned 3 groups ($n = 5, 6, 4$). Scale bar, 1 mm. **e, f** Representative

TRAP staining (**e**) and quantification of osteoclast surface per bone surface (Oc.S/BS), osteoclast number per bone perimeter (Oc.N/B.Pm), and osteoclast number per tissue area (Oc.N/T.Ar) (**f**) in tibial bone sections from the aforementioned 3 groups ($n = 5, 6, 4$). Triangles illustrate osteoclasts. Scale bar, 100 μm. Data are shown as mean ± SEM. Symbols represent individual mice. *P* values were determined by one-way ANOVA Tukey's test (7d, 7f) for multiple comparisons. Asterisks mark statistically significant difference (*$P < 0.05$, **$P < 0.01$, ****$P < 0.0001$). Source data are provided as a Source data file.

were randomly assigned to experimental groups, which were distributed among cages to compensate for covariates. In all experiments all measured values that met Quality Control criteria (e.g., correct genotype, induced disease model, effective staining, gene expression) were included and statistically analyzed. This exclusion criteria were predetermined. All experimental findings were reliably reproduced at least 3 times and mostly pooled. In some instances, representative experiments were displayed.

#### Eosinophil reconstitution
Nine weeks old ΔdblGATA mice were reconstituted with $3 \times 10^6$ eosinophils. The bone phenotype was assessed 14 days later.

#### IL-5 treatment
Nine weeks old ΔdblGATA mice were treated with intraperitoneal injection of 500 ng of recombinant murine IL-5 (BioLegend; Cat# 581506) daily for two consecutive weeks.

#### Ovariectomy (OVX)-induced osteoporosis
Twelve weeks old female mice were bilaterally ovariectomized and the ovaries of the sham group were left intact. Immediately post-surgery, mice were administered Buprenorphine for pain prevention. Regular monitoring was conducted to ensure there were no unforeseen health limitations. The OVX mice developed osteoporosis within 6 weeks and were sacrificed at an age of 18 weeks. To verify the successful removal

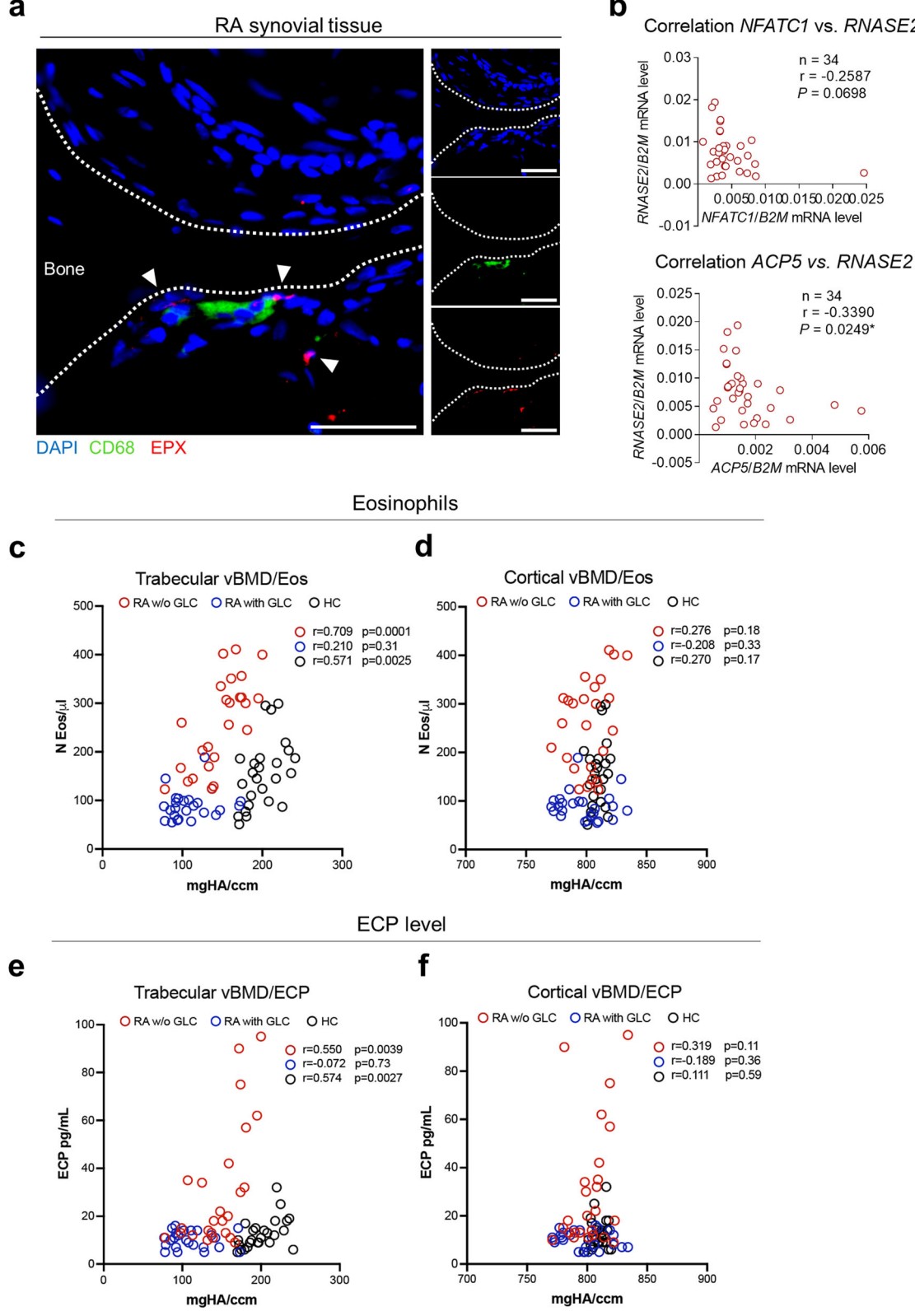

Eosinophils

ECP level

of the ovaries, the uterus weight was assessed at the end of the experiment.

**K/BxN serum transfer arthritis (STA)**
Nine weeks old female and male mice were injected intraperitoneally with 200 μl pooled serum from arthritic adult K/BxN mice as described[66]. K/BxN mice spontaneously develop autoimmune

arthritis independent of their sex[67]. Development of arthritis was evaluated for each paw using a semi-quantitative scoring system (0–4 per paw; maximum score of 16) as previously described[68]. Mice were analyzed at day 12 post-serum transfer. In addition, mice were injected intravenously with 40 μg of mouse eosinophil peroxidase (EPX) protein (Abbexa; Cat# ABX653285) in 0.9% NaCl as compared with vehicle control (0.9% NaCl) at day 0 and 4 of STA. Mice were

**Fig. 8 | Eosinophils and eosinophil cationic protein levels are correlated with bone mass in humans. a** Representative images of immunofluorescence microscopy of synovial tissue from rheumatoid arthritis (RA) patients. Poly-nucleated osteoclasts are stained with CD68 (green), eosinophils with EPX (red), and nuclei are visualized by DAPI (blue). Scale bar, 50 μm. Localization of eosinophils near osteoclasts in synovial tissue was detected independently in sections from 5 RA patients. **b** Correlation between the gene expression of eosinophil gene *RNASE2* with osteoclast-related genes *NFATC1* and *ACP5* in the blood of RA patients (*n* = 34). **c, d** Correlation of blood eosinophil counts with trabecular (**c**), and cortical (**d**) volumetric bone mineral density (vBMD) in RA patients without glucocorticoids (RA w/o GLC), with glucocorticoids (RA with GLC), and healthy controls (HC) (*n* = 25). **e, f** Correlation of serum eosinophil cationic protein (ECP) levels with trabecular (**e**), and cortical (**f**) vBMD in RA w/o GLC, RA with GLC, and HC (*n* = 25). Symbols represent individual participants. *P* values are based on one-tailed Spearman's correlation coefficients (8b, 8c, 8e). Asterisks mark statistically significant difference (**P* < 0.05, ***P* < 0.01). Source data are provided as a Source data file.

monitored every day to ensure there were no unforeseen health limitations.

## Micro-computed tomography imaging (μCT) in patients and controls

High-resolution peripheral quantitative computed tomography (HR-pQCT) imaging was performed by an Xtreme CT scanner (Scanco, Bruettiselien, Switzerland). Scan region was selected according to the manufacturer's standard in vivo protocol of the ultra-distal radius of the dominant hand. For measurement, the patients' hand was immobilized using a carbon fiber shell to reduce movement. Standardization of measurements was ensured by daily cross-calibrations with a standardized control phantom (Moehrendorf, Germany). The region of interest was determined with an anteroposterior scout view and was fixed 9.5 mm proximal from the reference line. The effective dose for each scan was lower than 3 μSv. The reference line was set manually. The scan region of interest was examined in 110 parallel slices (82 μm voxel size) with a total measurement time of 2.8 min. All measurements and evaluations were performed using the manufacturer's standard software. Motion grading (one to five) of each scan was performed using Scanco SOP scale, and scans graded higher than 3 were excluded from analysis. Three-dimensional volumetric bone mineral density (vBMD) of the total radius (Total BMD, mg hydroxyapatite/cm³), the cortical shell (Ct. BMD, mg HA/cm³) and the trabecular compartment (Tb.BMD, mg HA/cm³) were extracted. All these parameters were calculated by an automated software.

## Immunofluorescence staining with human synovial slides

Epitopes were retrieved from deparaffinized sections using a heat-induced method. As described before[19], sections were alternatively bathed in boiling citrate buffer (10 mM citric acid monohydrate, pH 6.0) and Tris-EDTA buffer (10 mM Tris base, 1 mM EDTA, 0.05% Tween-20, pH 9.0). Each bathing step was repeated five times for 2 min each. After permeabilization with Triton™ X-100 (MilliporeSigma; Cat# X100) and washing with 1x PBS, sections were blocked first for endogenous biotin with Endogenous Biotin-Blocking Kit (Thermo Fisher

Scientific; Cat# E21390) according to the manufacturer's instructions and then with 2.5% goat serum in 10 mM HEPES at RT for 1 h. Sections were incubated with primary antibody anti-EPX (clone AHE-1; mouse; 1:200; Abcam; Cat# ab190715) and anti-CD68 (clone FA-11; biotin; 1:100; GeneTex; Cat# GTX43914) in 2.5% goat serum in 10 mM HEPES overnight at 4 °C. After washing in 1x PBS, sections were incubated with the secondary antibodies AlexaFluor®555 goat anti-mouse IgG H&L (1:200; Abcam; Cat# ab150114) and AlexaFluor®488 Streptavidin (1:50; BioLegend; Cat# 405235) in 10 mM HEPES at RT for 3 h. After washing with 1x PBS, sections were mounted with Fluoroshield™ with DAPI (MilliporeSigma; Cat# F6057) and covered with coverslips. Images were acquired with the All-in-One Fluorescence Microscope BZ-X710 (KEYENCE).

## Micro-computed tomography imaging (μCT) of murine bones

Mouse tibias were fixed in 4% PFA/PBS (pH 7.4) overnight and transferred to 70% ethanol. All μCT images were taken with the cone-beam Desktop Micro Computer Tomograph μCT 40 (SCANCO Medical). The settings employed for measuring calcified tissue in murine bones were configured as follows: a kilovoltage (kVp) of 55, a current of 145 μA, an integration time of 200 ms for 500 projections per 180° rotation, and an isotropic voxel size of 6.0 μm. The assessment of the trabecular and cortex structure of the proximal tibia metaphysis involved selecting a volume of interest that began 420 μm (equivalent to 50 slices) from the center of the growth plate and extended 1680 μm (equivalent to 200 slices) towards the distal direction. The 3D-modeling of the bone was performed with optimized grayscale thresholds of the operating system Open VMS (SCANCO Medical) including the extraction of the bone parameters: bone volume per total volume (BV/TV), trabecular number (Tb.N), trabecular thickness (Tb.Th), and trabecular separation (Tb.Sp).

## Light sheet fluorescence microscopy (LSFM) with tibial bone

For eosinophil visualization in long bones, Cx3cr1creR26-tdTomato mice were injected intravenously with 5 μg/mouse of Alexa Fluor® 647 rat anti-mouse Siglec-F (clone E50-2440; BD Pharmingen; Cat# 562680) in 200 μl 1x PBS 1 h before sacrifice. After sacrifice, the mice were perfused with 15 mL 1x PBS with 5 mM EDTA through the left ventricle and afterwards with 15 mL 4% PFA/PBS (pH 7.4) to rinse erythrocytes and fix the bone tissues from inside, respectively. Thereupon, the tibial bone was fixed in 4% PFA/PBS (pH 7.4) at 4 °C for 4 h, dehydrated with increasing alcohol concentrations and cleared with ethyl cinnamate (MilliporeSigma; Cat# 112372) as described[69]. The LSFM imaging was performed with a LaVision BioTec Ultramicroscope (LaVision BioTec) with an Olympus MVX10 zoom microscope body (Olympus), a LaVision BioTec Laser Module, an Andor Neo sCMOS Camera with a pixel size of 6.5 μm and detection optics with an optical magnification range 1.263–12.63 and a numerical aperture (NA) of 0.5. A 488 nm optically pumped semiconductor laser (OPSL) was used for generation of autofluorescent signals. A 561 nm OPSL was used to detect tdTomato-positive osteoclasts. For Siglec-F-AF647 excitation, a 647 nm diode laser was used. Emitted wavelengths were detected with specific detection filters: 525/50 nm for autofluorescence, 620/60 nm for tdTomato and 680/30 nm for Siglec-F-AF647. An optical zoom factor of 2.5, a thickness of 3 μm, and a sheet NA of 4 μm was used.

## Table 1 | Patient's characteristics

| | RA (GLC–) | RA (GLC+) | HC |
|---|---|---|---|
| **Demographic characteristics** | | | |
| Age, years (mean ± SD) | 51.9 ± 9.5 | 50.6 ± 10.4 | 52.0 ± 10.6 |
| Sex, females (N/total) | 16/25 | 16/25 | 16/25 |
| BMI (mean ± SD) | 25.5 ± 3.7 | 25.4 ± 4.0 | 24.3 ± 3.1 |
| **Disease-specific characteristics** | | | |
| Disease duration, years (mean ± SD) | 1.54 ± 1.04 | 1.60 ± 1.15 | – |
| Disease activity score 28 (mean ± SD) | 3.18 ± 0.49 | 3.26 ± 0.44 | – |
| **Bone parameters** | | | |
| Trabecular BV, mgHA/ccm (mean ± SD) | 147 ± 31 | 109 ± 27 | 200 ± 21 |
| Cortical BV, mgHA/ccm (mean ± SD) | 802 ± 15 | 797 ± 18 | 810 ± 6 |
| **Eosinophil parameters** | | | |
| Eosinophil counts, cell/μl (mean ± SD) | 258 ± 93 | 89 ± 29 | 155 ± 69 |
| ECP level, pg/mL (mean ± SD) | 30.5 ± 25.7 | 10.2 ± 3.5 | 12.6 ± 6.3 |

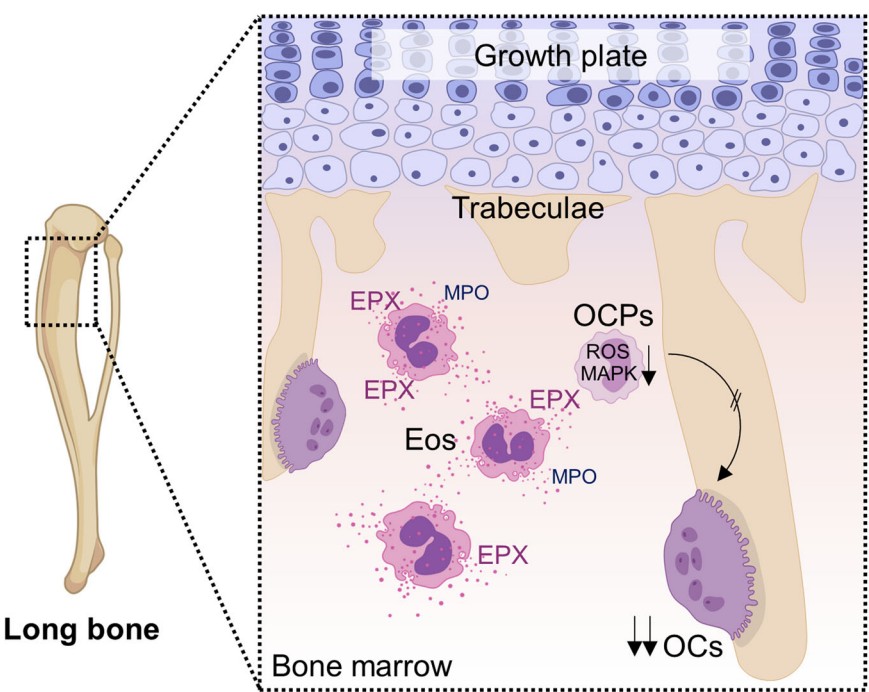

**Fig. 9 | Schematic overview of eosinophil-dependent osteoclast inhibition.** Eosinophils (Eos) are localized in proximity to bone-resorbing osteoclasts (OCs) in the bone marrow. Eosinophils release eosinophil peroxidase (EPX) and reduce the level of reactive oxygen species (ROS) and the phosporylation of mitogen-activated protein kinases (MAPKs) in osteoclast precursors (OCPs). In this way, eosinophils influence the homeostasis of bone and reduce pathological bone loss. Figure partially generated with biorender.com.

Evaluations were done with Image J and Imaris software (Oxford Instruments).

### Cryo-sectioning of tibial bone
To generate cryo-sections of tibial bone from Cx3cr1$^{cre}$R26-tdTomato mice, bones were fixed in 4% PFA/PBS (pH 7.4) for 12 h at 4 °C, incubated for 10 days in decalcification buffer (14% EDTA free acid, NH$_4$OH, pH 7.2) and embedded in OCT Compound (Sakura Finetek; Thermo Fisher Scientific; Cat# 12351753). Seven-μm-thick histological sections were generated with the cryostat Leica CM 3050 S (Leica Biosystems).

### Confocal laser scanning microscopy (CLSM)
Cryo-sections of tibial bone from Cx3cr1$^{cre}$R26-tdTomato mice were blocked in 2.5% goat serum in 10 mM HEPES and permeabilized with 0.1% saponin in PBS at RT for 1 h. Sections were incubated with primary antibody anti-EPX (polyclonal; rabbit; 1:100; Abcam; Cat# ab65319) and Alexa Fluor® 647 anti-mouse/rat CD61/β3 integrin (clone 2C9.G2; 1:100; BioLegend; Cat# 104314) in 2.5% goat serum in 10 mM HEPES overnight at 4 °C. After washing in 1x PBS, sections were incubated with the secondary antibody AlexaFluor®488 goat anti-rabbit IgG (H + L) (1:50; Invitrogen; Cat# A-11008) in 10 mM HEPES at RT for 3 h. After washing with 1x PBS, sections were mounted with Fluoroshield™ with DAPI (MilliporeSigma; Cat# F6057) and covered with coverslips. Images were acquired with a Leica TCS SP 5 II confocal microscope (Leica Microsystems) with acousto-optic tunable filter and acousto-optical beam splitter, and equipped with photomultiplier tubes (PMTs) and hybrid detectors (HyD) on a DMI6000 CS frame. Representative images were generated with an HCX PL APO CS 63.0×1.30 GLYC 21 °C UV objective. Fluorescence signals were generated via three sequential scans. In the first imaging sequence DAPI and AF647 were simultaneously excited with a 458 nm argon laser and a 633 nm helium−neon laser, respectively. DAPI was detected with PMT at 470−520 nm and AF647 signals were detected with PMT at 650−750 nm. The second sequence for detecting Alexa Fluor 488 used an argon laser at 488 nm for excitation and a HyD detector at 500−560 nm. A third imaging

sequence involved an excitation of tdTomato with a 561 nm diode-pumped solid-state laser and its detection with a HyD detector at 600−650 nm. Generated images were processed with Imaris software (Oxford Instruments).

### Histological analysis
Tibiae of mice were fixed overnight in 4% PFA/PBS (pH 7.4) and afterwards decalcified (14% EDTA free acid, NH$_4$OH, pH 7.2) for 14 days until bones were pliable. Serial paraffin sections (2 μm) were stained with tartrate-resistant acid phosphatase (TRAP) staining with the acid phosphatase, leukocyte (TRAP) kit (MilliporeSigma; Cat# 387A) for quantifying bone mass and osteoclast numbers. The quantification of bone volume per total volume (BV/TV), trabecular number (Tb.N), trabecular thickness (Tb.Th), trabecular separation (Tb.Sp), osteoclast surface per bone surface (Oc.S/BS), osteoclast number per bone perimeter (Oc.N/B.Pm), and osteoclast number per tissue area (Oc.N/T.Ar) was performed on an Axio Lab.A1 microscope (Carl Zeiss), equipped with a digital camera and image analysis system (Osteo-Measure, Osteometrics).

### Flow cytometry
For cell isolation from the bone marrow (BM), the left femur was flushed 3x with 1x PBS. Then, the BM cells were lysed with 3 mL RCL buffer and put through 70 μm cell strainers. Blood was lysed twice with 5 mL RCL buffer and put through 70 μm cell strainer. For surface marker staining, isolated single cells were incubated with the following antibodies: anti-CD45 APC-eFluor™ 780 (clone 30-F11; 1:800; eBioscience; Cat# 47-0451-82), anti-CD45 PerCP/Cyanine 5.5 (clone 30-F11; 1:400; BioLegend; Cat# 103132), anti-CD11b FITC (clone M1/70; 1:400; BD Pharmingen; Cat# 557396), anti-CD11b PerCP (clone M1/70; 1:400; BioLegend; Cat# 101230), anti-CD11b Brilliant Violet 605™ (M1/70; 1:400; BioLegend; Cat# 101257), anti-Ly6G PerCP/Cyanine 5.5 (clone 1A8; 1:800; BioLegend; Cat# 127616); anti-Ly6C PE/Cyanine 7 (clone HK1.4; 1:1000; BioLegend; Cat# 128018), anti-CD115 (CSF-1R) Brilliant Violet 421™ (clone AFS98; 1:800; BioLegend; Cat# 135513), anti-Siglec-F

PE (clone E50-2440; 1:400; BD Pharmingen; Cat# 562068), anti-CD117 (c-Kit) Pacific Blue™ (clone 2B8; 1:400; BioLegend; Cat# 105820), anti-CD117 (c-Kit) Brilliant Violet 605™ (clone 2B8; 1:800; BioLegend; Cat# 105847), anti-CD49b PE/Cyanine 7 (clone DX5; 1:400; BioLegend; Cat# 108922), anti-FcεRIα PE/Cyanine 7 (clone MAR-1; 1:100; BioLegend; Cat# 134318), anti-CD200R3 PE (clone Ba13; 1:400; BioLegend; Cat# 142206), anti-CD3ε PE/Cyanine5 (clone 145-2C11; 1:400; BioLegend; Cat# 100310), anti-CD3ε APC (clone 145-2C11; 1:400; BioLegend; Cat# 100312), anti-CD19 APC (clone 1D3; 1:400; BD Pharmingen; Cat# 550992), anti-CD335 (NKp46) APC (clone 29A1.4; 1:400; BioLegend; Cat# 137608), anti-CD31 Alexa Fluor® 647 (clone MEC13.3; 1:400; BioLegend; Cat# 102516), anti-CD61 PE (clone 2C9.G2; 1:400; BioLegend; Cat# 104308), anti-CD41 Pacific Blue™ (clone MWReg30; 1:400; BioLegend; Cat# 133932), anti-Ly-6A/E (Sca-1) PE/Cyanine 7 (clone E13-161.7; 1:400; BioLegend; Cat# 122514), anti-CD16/CD32 APC (clone 93; 1:400; eBioscience; Cat# 17-0161-82), anti-CD34 FITC (clone RAM34; 1:400; BD Pharmingen; Cat# 560238), anti-CD45R/B220 PerCP (clone RA3-6B2; BD Pharmingen; Cat# 553093), anti-CD45R/B220 Brilliant Violet 510™ (clone RA3-6B2; 1:400; BioLegend; Cat# 103247), anti-TER-119 APC-efluor™ 780 (clone TER-119; 1:400; eBioscience; Cat# 47-5921-82), anti-CD71 FITC (clone RI7217; 1:400; BioLegend; Cat# 113806), anti-CD5 PerCP (clone 53-7.3; 1:200; BD Pharmingen; Cat# 553025) in 1x PBS in the dark at 4 °C for 20 min. After washing, cells were resuspended in FACS buffer (1x PBS with 2% FBS and 2 mM EDTA) for flow cytometric analyses. Flow cytometry was performed on the Gallios™ or Cytoflex S flow cytometer (all from Beckman Counter). Flow cytometry data were analyzed by Kaluza 2.1 (Beckman Counter) or FlowJo (BD Biosciences).

**Eosinophil sorting for in vitro analysis**
Eosinophils were sorted from the blood and BM of IL-5tg/4get mice as described previously[19]. The blood was lysed twice with 10 mL RCL buffer. BM cells were flushed out from the femur and tibia. Cells from blood and BM were pooled and put through 70 μm cell strainers. Cells were stained with anti-CD45 APC-eFluor™ 780 (clone 30-F11; 1:800; eBioscience; Cat# 47-0451-82), anti-CD125 (IL-5Rα) APC (clone REA343; 1:100; Miltenyi Biotec; Cat# 130-118-561), and anti-Siglec-F PE (clone E50-2440; 1:400; BD Pharmingen; Cat# 562068) in 1x PBS for 20 min in the dark at 4 °C. After washing, cells were resuspended in FACS buffer for sorting on MoFlo Astrios EQ (Beckman Counter). CD45$^+$ CD125$^{int}$ Siglec-F$^+$ granulocytes were sorted into 1x PBS with 2% FBS for in vitro experiments. Besides, $2 \times 10^6$/mL sorted eosinophils were cultured for 48 h in αMEM and GlutaMAX (Gibco; Cat# 32571028) with 10% FBS (Gibco; Cat# A5256701) and 1% penicillin/streptomycin (Gibco; Cat# 15140122) at 37 °C and 5.5% CO$_2$ to generate eosinophil supernatant. After culture, the supernatant was separated from the cells by centrifugation and stored at −80 °C. The viability of the cultured eosinophils after 48 h was analyzed by flow cytometry with DAPI (Roche; Cat# 10236276001).

**Negative selection of neutrophils by magnetic cell separation**
Neutrophils were isolated from the blood and BM of BALB/c WT mice by untouched magnetic cell separation (MACS) with the Neutrophil Isolation Kit (Miltenyi Biotec; Cat# 130-097-658), following the manufacturer's instructions. The purity and viability of neutrophils before and after MACS was analyzed by flow cytometry with anti-CD11b FITC (clone M1/70; 1:400; BD Pharmingen; Cat# 557396), anti-Ly6G PerCP/Cyanine 5.5 (clone 1A8; 1:800; BioLegend; Cat# 127616), and DAPI (Roche; Cat# 10236276001).

**Proteome profiler assay**
The Proteome Profiler Array was performed with the Mouse XL Cytokine Array Kit (R&D; Cat# ARY028) according to manufacturer's instructions with 1 mL of eosinophil supernatant. The array membrane

was exposed to an X-ray film for 10 min. The integrated pixel density was quantified with ImageJ.

**Osteoclast differentiation**
Total BM cells from BALB/c WT mice were isolated by flushing the femur and tibia. The cells were plated overnight at 37 °C and 5.5% CO$_2$ in a 100 × 20 mm dish in osteoclast medium, composed of αMEM and GlutaMAX (Gibco; Cat# 32571028) with 10% FBS (Gibco; Cat# A5256701) and 1% penicillin/streptomycin (Gibco; Cat# 15140122), supplemented with 5 ng/mL M-CSF (PeproTech; Cat# 315-02). The next day (day 0), nonadherent bone marrow derived monocytes (BMMs) were collected, washed, and further cultured in osteoclast medium with 20 ng/mL M-CSF and 10 ng/mL RANKL (PeproTech; Cat# 315-11) in 96-well plates (TRAP; 200 μL/well) or 48-well plates (RNA; 500 μL/well) at the concentration of $1 \times 10^6$ cells/mL at 37 °C and 5.5% CO$_2$. In addition, BMMs were stimulated on day 0 of culture for 48 h with the following stimulants: eosinophils (1:2, 1:1, 2:1, 4:1 Eos/BMMs ratio), neutrophils (1:1 Neutros/BMMs ratio) eosinophil supernatant (1:4, 1:3, 1:2 dilution; M-CSF and RANKL level was adjusted to 20 ng/mL and 10 ng/mL, respectively), BMM supernatant (1:2 dilution; M-CSF and RANKL level was adjusted to 20 ng/mL and 10 ng/mL, respectively), 10 ng/mL recombinant mouse IL-4 (BioLegend; Cat# 574304), 0.1–0.8 μM recombinant mouse cystatin C protein (Novus Biologicals; Cat# NBP25959050), 5–10 μg/mL recombinant mouse myeloperoxidase (MPO) protein (R&D Systems; Cat# 3667-MP), 0.05-1 μg/mL mouse EPX protein (Abbexa; Cat# ABX653285), 10 μM MPO/EPX inhibitor (MilliporeSigma; C$_7$H$_9$N$_3$O; Cat# 475944), 0.1 μM TGF-β receptor inhibitor Galunisertib (MilliporeSigma; Cat# SML2851). In addition, BMMs were stimulated on day 0 of culture for 48 h with eosinophils (1:1 Eos/BMMs ratio) that were separated by a membrane with a pore size of 1 μm (THINCERT; Greiner Bio-One; Cat# 662610). The viability of BMMs, detached from the plates by accutase (MilliporeSigma; Cat# A6964), was analyzed 48 h after culture by flow cytometry with DAPI (Roche; Cat# 10236276001). Otherwise, the medium was changed on day 2 and additionally on day 4 if necessary. In some conditions, the stimulation of BMMs started at day 2 of differentiation after the first medium change. Fully differentiated osteoclasts (day 3–5) were either processed for RNA or washed with PBS, fixed with fixation buffer, and stained with TRAP solution. Images were acquired with the All-in-One Fluorescence Microscope BZ-X710 (KEYENCE) and quantification of osteoclast numbers was performed with ImageJ.

**Demineralization assay**
For the demineralization assays, $1 \times 10^6$ BMMs/mL were cultured in osteoclast medium with supplements on 24-well plates coated with hydroxyapatite (Corning; Cat# 3987) at 37 °C and 5.5% CO$_2$. BMMs were stimulated at day 0 of culture for 48 h with eosinophils (1:1 Eos/BMMs ratio) and eosinophil supernatant (1:2 dilution) compared with unstimulated control. The medium was changed at day 2. After 3 days of differentiation, osteoclasts on the hydroxyapatite-coated plates were lysed with dH$_2$O and the plates were incubated with 5% sodium hypochlorite (MilliporeSigma; Cat# 425044) for 5 min and stained with Von Kossa staining. Images of the whole well were acquired with the All-in-One Fluorescence Microscope BZ-X710 (KEYENCE) and percentage of the demineralized area was quantified with ImageJ.

**Bulk RNA-sequencing with osteoclasts**
BMMs from BALB/c WT mice were stimulated on day 0 of culture for 48 h with eosinophils (2:1 Eos/BMMs ratio; OCsEo) or eosinophil supernatant (1:2 dilution; OCsEoS) compared with unstimulated control (OCsCtr) in osteoclast medium with supplements at 37 °C and 5.5% CO$_2$. At day 2, the cells were washed and medium was changed. Total osteoclast RNA was harvested on day 3 of culture and purified with the RNeasy Mini Kit (QIAGEN; Cat# 74104) according to the

**Table 2 | Murine and human qPCR primers forward and reverse**

| Gene | Forward | Reverse |
|---|---|---|
| *Actb murine* | 5'TGTCCACCTTCCAGCAGATGT3' | 5'AGCTCAGTAACAGTCCGCCTAGA3' |
| *Acp5 murine* | 5'CGACCATTGTTAGCCACATACG3' | 5'TCGTCCTGAAGATACTGCAGGTT3' |
| *ACP5 human* | 5'TGAGGACGTATTCTCTGACCG3' | 5'CACATTGGTCTGTGGGATCTTG3' |
| *B2M human* | 5'GATGAGTATGCCTGCCGTGTG3' | 5'CAATCCAAATGCGGCATCT3' |
| *Ctsk murine* | 5'AGGGCCAACTCAAGAAGAAAACT3' | 5'TGCCATAGCCCACCACCAACACT3' |
| *Epx murine* | 5'CTGTCTCCTGACTAACCGCTCT3' | 5'TCAGCGGCTAGGCGATTGTGTT3' |
| *Mmp9 murine* | 5'GCTGACTACGATAAGGACGGCA3' | 5' TAGTGGTGCAGGCAGAGTAGGA3' |
| *Nfatc1 murine* | 5'GGTGCCTTTTGCGAGCAGTATC3' | 5'CGTATGGACCAGAATGTGACGG3' |
| *NFATC1 human* | 5'GTCCTGTCTGGCCACAAC3' | 5'GGTCAGTTTTCGCTTCCATC3' |
| *RNASE2 human* | 5'AGCGCGGAGACTGGGAAAC3' | 5'TTCAAACCATTGAGCCCAGGT3' |
| *Tnfrsf11a murine* | 5'TTGTGGCAGGGGACTTTAAC3' | 5'ATTGTCATCCTGCCCTCAAC3' |
| *Traf6 murine* | 5'AAAGCGAGAGATTCTTTCCCTG3' | 5'ACTGGGGACAATTCACTAGAGC3' |

manufacturer's instructions. Bulk RNA-seq. was carried out by Novogene (UK). As described by novogene: A total amount of 1 μg RNA per sample was used as input material for the RNA sample preparations. Sequencing libraries were generated using NEBNext® Ultra™ RNA Library Prep Kit for Illumina® (NEB, Cat# E7770) following manufacturer's recommendations and index codes were added to attribute sequences to each sample. The clustering of the index-coded samples was performed according to the manufacturer's instructions. After cluster generation, the library preparations were sequenced on an Illumina platform and paired-end reads were generated. Clean reads were obtained by removing reads containing adapter, reads containing poly-N, and low-quality reads from raw data. All the downstream analyses were based on the clean data with high quality. The filtered reads were aligned to the reference genome *Mus musculus* GRCm38/mm10 [https://www.ncbi.nlm.nih.gov/datasets/genome/GCF_000001635.20/] with Hisat2 v2.0.5. Feature Counts v1.5.0-p3 was used to count the reads numbers mapped to each gene. All original RNA-seq data (fastq files) and the read counts have been deposited in the Gene Expression Omnibus (GEO) database under the accession code GSE203541. Principal component analysis and volcano plots were generated with the R packages glmpca (version 0.2.0) and EnhancedVolcano (version 1.12.0), respectively. Differential expression analysis between 2 conditions or groups (4 biological replicates per condition) was performed using DESeq2 R package (version 1.34.0) with log2 fold change cut off 0.5. The resulting $P$ values were adjusted using the Benjamini–Hochberg approach for controlling the false discovery rate (FDR). Genes with an adjusted $P$ value ($P_{adj}$) less than 0.05 were assigned as differentially expressed.

## Mitochondrial reactive oxygen species (ROS) measurement

BMMs from BALB/c WT mice were cultured in $1 \times 10^6$ cells/well in 1 mL/well osteoclast medium with supplements on 24-well plates at 37 °C and 5.5% $CO_2$. BMMs were stimulated at day 0 of culture for 48 h with eosinophils (1:1 Eos/BMMs ratio), eosinophil supernatant (1:2 dilution), 10 μg/mL of MPO, and 0.05–0.1 μg/mL of EPX compared with unstimulated control. At day 2, the cells were detached from the plate by accutase and stained with MitoSOX™ Red (Invitrogen; Cat# M36008) according to the manufacturer's instructions. Afterwards, the cells were incubated with the following antibodies: anti-CD45 APC-eFluor™ 780 (clone 30-F11; 1:800; eBioscience; Cat# 47-0451-82), anti-CD11b FITC (clone M1/70; 1:400; BD Pharmingen; Cat# 557396), and anti-CD115 (CSF-1R) Brilliant Violet 421™ (clone AFS98; 1:800; BioLegend; Cat# 135513) in 1x PBS in the dark at 4 °C for 20 min. After washing, cells were resuspended in FACS buffer (1x PBS with 2% FBS and 5 mM EDTA) for flow cytometric analyses. Flow cytometry was performed on the Gallios™ flow cytometer (Beckman Counter). Flow cytometry data were analyzed by Kaluza 2.1 (Beckman Counter).

## MULTI-SPOT MAP kinase phosphoprotein assay

For the quantification of phospho-JNK (Thr183/Tyr185), phospho-p38 (Thr180/Tyr182), and phospho-ERK1/2 (Thr/Tyr: 202/204; 185/187), $5 \times 10^5$ BMMs/well were cultured in 48-well plates in 500 μL/well osteoclast medium with supplements at 37 °C and 5.5% $CO_2$. BMMs were stimulated at day 0 of culture for 48 h with eosinophils (1:1 Eos/BMMs ratio), eosinophil supernatant (1:2 dilution), and 1 μg/mL of EPX compared with unstimulated control. The medium was changed on day 2. After 3 days of differentiation, the MAP Kinase Whole Cell Lysate Kit (MESO SCALE DIAGNOSTICS; Cat# K15101D) was performed according to the manufacturer's instructions. Briefly, osteoclasts were lysed with 50 μL/well of freshly prepared protein lysis buffer. Protein concentration was measured with DC Protein Assay Kit (Bio-Rad; Cat# 5000111) according to the manufacturer's instructions. 10 μg protein lysate/well was applied in the MULTI-SPOT MAP Kinase Phosphoprotein Assay. Finally, the signal intensity of phospho-JNK, phospho-p38, and phospho-ERK-1/2 was measured with the MSD SECTOR imager.

## Enzyme-linked immunosorbent assay (ELISA)

Eosinophil supernatant was used for the measurement of cystatin C, MPO, and EPX protein level. Levels of cystatin C and MPO were measured by ELISA using mouse cystatin C/MPO DuoSet ELISA kit (all from R&D; Cat# DY1238; Cat# DY3667) according to the manufacturer's instructions. EPX was measured with the Mouse EPX ELISA Kit (MyBiosource; Cat# MBS2706597) according to the manufacturer's instructions. TGFβ level in the supernatant of eosinophils and BMMs was detected with the mouse TGF-beta 1 DuoSet ELISA (R&D; Cat# DY1679). Serum levels of human eosinophil cationic protein (ECP) were assessed by commercial ELISA (Cusabio; Cat# CSB-E11729h).

## RNA isolation and real-time PCR

Murine RNA from in vitro differentiated osteoclasts or femoral bones was extracted using RNA-Solv® Reagenz (VWR Peqlab; Cat# R6830-02) according to the manufacturer's instructions. Mouse femurs were first homogenized using Precellys® Steel kit (VWR Bertin corp.; Cat# P000910-LYSK0-A) on Precellys 24 (VWR Peqlab). The blood of RA patients was collected in PAXgene Blood RNA Tubes (QIAGEN; Cat# 762165), and the RNA was isolated using the PAXgene Blood RNA Kit (QIAGEN; Cat# 762174) according to the manufacturer's instructions. Extracted RNA was freed from genomic DNA using DNase I kit (Thermo Fisher Scientific; Cat# EN0521) and reversely transcribed into cDNA using high capacity cDNA Reverse Transcription Kit (AppliedBiosystems; Cat# 4368814). Real-time PCR was performed using Takyon ROX

SYBR 2X MasterMix dTTP blue (Eurogentec; Cat# UF-RSMT-B0701) on CFX96TM Real-Time System (Bio-Rad) with the primers listed in Table 2. Gene expression was normalized with *Actb* for mouse and *B2M* for human.

## Statistics

All statistical analyses were performed using Graph-Pad Prism-Software 9. Data was presented as mean +/− standard error of mean (SEM). Normal distribution of the samples was tested with D'Agostino & Pearson test for $n \geq 8$ and Shapiro−Wilk test for $n < 8$. Statistical significance was calculated by unpaired two-tailed $t$ test (normally distributed variables) or two-tailed Mann−Whitney test (non-normal variables) for single comparisons. For multiple comparisons, one-way ANOVA Dunnett's test, one-way ANOVA Tukey's test or two-way ANOVA Tukey's test was used as parametric test and Kruskal-Wallis Dunn's test as non-parametric test. Human correlation analyses were statistically testes with Spearman's correlation coefficients. Statistical details (e.g., number of animals/participants per group, number of independent experiments, etc.) can be found in the figure legends. $P$ values less than 0.05 were considered statistically significant.

## Reporting summary

Further information on research design is available in the Nature Portfolio Reporting Summary linked to this article.

## Data availability

The bulk RNA-sequencing data generated in this study have been deposited in the Gene Expression Omnibus (GEO) database under accession code "GSE203541". Source data are provided with this paper.

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

## Acknowledgements

We thank Daniela Weidner, Franceska Jelas, Jule Lindörfer, Barbara Happich, and Nicole Berndt for great technical assistance. The authors thank Dr. Wolfgang Baum for the generation of K/BxN serum. We thank Uwe Appelt and Markus Mroz of the Core Unit Cell Sorting and Immunomonitoring facility for their technical support in flow cytometry and for cell sorting. We want to thank the Optical Imaging Centre Erlangen (OICE) within the Z project of the CRC 1181 for support with confocal microscopy. In addition, the authors thank Koray Tascilar for providing valuable guidance on the statistical analysis of the experimental data. This study was supported by the Interdisciplinary Center for Clinical Research grant J90 (D.A.); J76 (K.K.); and A77 (A.B.), the Collaborative Research Center 1181 project A01 (A.B., G.S.); A02 (D.V.); and A05 (S.F.), the Collaborative Research Center/Transregio 369 project B03 (D.A.); A02 (A.B.); and B05 (A.B.); the German Research Foundation grant FOR2886 TP02 (A.B.), and the European Research Council Consolidator Grant LS4-ODE (A.B.), and the Synergy Grant 4D Nanoscope (G.S.).

## Author contributions

D.A. acquired funding, designed and performed experiments, interpreted results, and wrote the manuscript. K.K., M.L., and Z.C. performed experiments and collected data. B.K. and G.K. offered expertise in osteoclast imaging and provided Cx3cr1^creR26-tdTomato mice. M.R. and S.F. provided experimental material. D.V. offered expertise in eosinophil

research and provided GEM mice. G.S. and A.B. acquired funding, designed the study and experiments, interpreted results, and wrote the manuscript. All authors read and commented on the manuscript.

## Funding

## Competing interests
The authors declare no competing interests.
