## [Peer Review file · Nature Communications]

REVIEWER COMMENTS

Reviewer #1 (Remarks to the Author):

In this study Andreev D and colleagues report that eosinophils promote bone density via inhibiting osteoclast formation and activity via one of its granule proteins, eosinophil peroxidase, using a variety of assays including a murine models of osteoporosis and serum transfer arthritis, ex vivo culture studies, histopathology and imaging of trabecular bone. Hyper-eosinophilia through the overexpression of IL-5 rescued this phenotype of increased pathology bone loss in the ovariectomy model. During the serum transfer arthritis model, the addition of two doses of EPX rescued WT mice from severe inflammatory bone loss suggesting that eosinophil-derived EPX may be the soluble factor inhibiting osteoclast activity in mice. Human data takes the form of synovial tissue biopsies, peripheral blood transcriptomics and blood eosinophil counts in rheumatoid arthritis patients versus healthy volunteers, correlating them with bone mass as measured by mCT.

If true, these findings would have significant medical implications because anti-eosinophil therapies are routinely being used for treatment of a variety of chronic conditions including asthma, EGPA, hypereosinophilic syndromes and EGIDs, with the hope of avoiding the toxic side effects of conventional therapies such as corticosteroids. While the data appear compelling, there are a number of major concerns which makes the results and conclusions less convincing.

Major Concerns (in no particular order)

1. It is unclear the relevance of measuring peripheral blood derived mRNA of osteoclast genes in human subjects– shouldn't these gene transcripts be in the bone only?
2. Details of concurrent or prior medications are not described for the patients. Steroids, methotrexate, are common drugs used to treat RA and have effects on numbers of eosinophils. Those who have mild RA may be on fewer medications or not be on meds at all. This would create a confounding element to the correlations reported in Figure 7. In fact, since long term corticosteroids use would decrease bone mineral density (and reduce eosinophil counts in the blood), this would be a significant confounder that would need to be accounted for.
3. Why was RA chosen to be studied? It is not typically thought to be an eosinophil-mediated disorder. Why not directly study patients with osteoporosis?
4. Age and gender are significant variables in bone mineral density and have an interaction too. A more complex statistical analysis should be done than just comparing mean ages.
5. Is mCT acceptable for measuring bone mineral density in human? Why not obtain DEXA scan results which is typically what clinicians use?

6. mCT data of WT or untreated murine controls seem to be different from experiment to experiment, with the spread being as wide as some of the significant differences between treatment and no treatment. Please comment on the expected degree and range of bone loss in these models.

7. Purifying eosinophils using positive selection via Siglec-F - these isolated eosinophils would be expected to die and extrude all sorts of non-specific proteins in this process, which could inhibit any cell growth. Would be useful to repeat with another cell type (neutrophils?) and see if their supernatants do the same – since PMNs don't make EPX. Alternatively, a blocking antibody to EPX would be a useful control to assess specificity as suggested. Or use negative selection to purify eosinophils.

8. Seeing eosinophil related genes in the bulk-RNA seq data set is concerning for residual eosinophil contamination in the bulk RNA from these osteoclast cultures. Perhaps those eosinophil genes can be masked/removed from analysis, and the PCA be repeated to see if grouping still occurs.

9. Since mCT data of trabecular bone is the main rodent clinical symptom, details on the following should be addressed:

1. Where on the tibia is the imaging acquired? Is a uniform size/volume acquired per sample? Best practices seem to image both axial and appendicular skeleton.

2. Comment on the choice of age of mice in this study. Given peak bone mineral density is at 16 weeks, trabecular bone decreases at weaning while cortical bone volume can continue to increase up to 7-12 months of age.

3. Some mouse strains have lower trabecular volume specifically -e.g. C57BL/6 and specifically in the tibia – was this accounted for in the analysis between different mouse backgrounds?

4. How do these magnitude of changes compare to other models of osteoporosis in mice?

10. Figure 1: Were male or female mice used for these experiments?

11. Figure 1 D and E: Δ dblGATA mice show significantly decreased bone mass compared to wild type controls. How do the authors reconcile this observation with published data by Kacena et al. in the Journal of Bone and Mineral Research that show increase in trabecular bone volume in GATA-1 deficient mice. Does eosinophil reconstitution of Δ dblGATA mice restore bone homeostasis?

12. The IL-5 transgenic mouse expresses supraphysiological levels of IL-5 which not only affects the number of blood eosinophils in the mouse, but likely other cells that have the IL-5R in mice (e.g. basophils, mast cells perhaps, and certain B cell subsets). If the experiments are not done to address whether it is an eosinophil specific effect that leads to increased bone mineral density (e.g. crossing the IL-5tg with an eosinophil knockout to show that the protective effect is abrogated to show that it is eosinophil specific and not supported by IL-5 on non-eosinophil cells), there needs to be some discussion of this point. Lastly, it should be noted that human B cells do not express the IL-5R which makes murine models utilizing IL-5 transgenes a little more challenging to compare directly with people.

13. Figure 2 H: Any LSFM imaging to support the statement in the text that "OC's likely internalize the Eos"

14. Figure 3 A and B: LIF and TGFb both are produced by eosinophils and have important roles in bone homeostasis. Was TGFb protein measured?

Reviewer #2 (Remarks to the Author):

Andreev and colleagues have done a tremendous effort to collect all the data presented in this manuscript. I am impressed and I would like to congratulate the authors on a well-written, comprehensive and complex manuscript - but admittedly also complicated dataset, which I feel is not always as clear as the authors state in the manuscript. This may simply be because I have overlooked something or simply did not understand it correctly. In this case, I would appreciate the authors if they could enlighten me, but maybe also consider if a potential future reader could have the same problems or thoughts. Therefore, I hope the authors will see my review in the right spirit since I am positive towards the manuscript although I am also skeptic.

To sum up my skepticism it is foremost based on what I see as a potential conflict between the in vivo and in vitro data. Although the in vivo effects are significant they are also modest in all experimental setups (when you look at the graphs). While the in vitro effects are very strong. Some of the in vitro effects I am also not totally convinced are direct effects of Eos. Below I will list a series of comments and requests. They are hard to group, but I hope the authors will find their way and recognize what my concerns are:

- Throughout the entire manuscript please modify statements that are overstatements such as: Do the Eos really CONTROL OCs in vivo. The differences are not convincing enough for such a strong statement
- Page 6 bottom: does the use of an inhibitor prove anything? Or does it maybe suggest? No inhibitor is 100% effective or specific...
- Page 8 title: do Eos really protect? No - maybe reduce but not protect!
- Page 9 center: verify – no the results do not verify and they certainly do not show that Eos control human bone homeostasis. What is shown are correlations – no more and no less.
- Page 2 bottom: Is TRAP and MMP9 crucial for osteoclastic bone resorption?
- There are many such exaggerations – please correct and please be thorough!
- What is the purity of the Eos cultures? 86%? And how about after 48h culture? I feel the authors do not consider well enough the potential consequences of contaminating cells.

- Is the FBS used for Eos and OCs the same?
- In cell cultures how are the control conditions treated? What would be a suitable control for e.g. the co-culture experiment? I find that is hard to answer, but it is critical to interpret the result correctly.
- Regarding the co-culture effects, I am sceptic that the effects observed are not just a density problem. The preOCs may simply not find each other? Would that not be observed when co-culturing with many other cell type?
- In conditioned media experiments what is added to the control cells? Is fresh media added here or was this also incubated for 48h prior to addition?
- Quantifications of OCs are consistently shown as more than or equal to 5 nuclei. This needs to be more than or equal to 3 nuclei.
- Fig. 1h, 1n and SF2: Isn't it puzzling that Acp5 increases dramatically and Ctsk is unchanged in Fig. 1h? Maybe this explains the modest in vivo effects on bone? In Fig. 1n Acp5 expression is lost but Ctsk is not significantly reduced? SF2 here the in vivo bone effect seems modest, but Ctsk is sky high? Does this add up and does this support the title and conclusions throughout the manuscript?
- The authors chose not to use bone as a substrate to test the activity of OCs in vitro. Therefore, the authors can also not make any comparison to in vivo effects since mouse bone does not just consist of hydroxyapatite but also of large amounts of collagen – and this is a critical difference. I would strongly urge the authors to use a bone substrate! The other substrate is convenient to use – but not even close to the natural substrate.
- How were the images chosen to use for quantification of mineral resorption? And how many pictures from each well?
- Does dbiGATA only affect the presence of Eos? How about other cell types? Are other cells in the bone marrow affected directly or indirectly? Could this potentially affect interpretations of the data?
- Do not show linear correlation lines when doing Spearman analyses! I assume you have chosen a Spearman correlation because there is no linear correlation. Especially in F7e and f they are far from linear – actually the correlations are not very convincing although significant.
- Fig. 1 – what does it mean for cells to be close in the bone marrow? The bone marrow cavity of a mouse is tiny. It is essential that the authors measures the distances between Eos and OCs in sections when coming with such a statement – images do not support this statement alone. How about direct contacts? Also all the in vitro results only investigate effects on OCs differentiation but not on mature OCs – so is it really relevant whether they are close to mature OCs?
- How could the authors show that it is truly a soluble factor from Eos that cause an effect on preOCs? Is the use of conditioned media enough? In this context it is especially important to consider how the control is treated?! And from what I can read in the methods, I am uncertain about this.
- Page 5 bottom: the actin rings in Fig. 2i are not convincing as having anything to do with active resorption – the pictures are simply not useful. I would also like to remind the authors that they have chosen not to use bone so the OCs is simply sitting on glass where such sealing zone structures does not look like on bone. Please revise carefully.

- Statistics: Statistical tests are parametric tests despite that in some experiments $n=4$?! I insist that the authors are very critical towards which tests they use and when. Have the authors checked that all datasets meet the requirements for doing the tests chosen? From the statistics paragraph, it does not seem so. Please go carefully through all tests used for all data sets and make sure that there are statistically sound arguments to use the chosen test.

REVIEWER COMMENTS

Reviewer #1 (Remarks to the Author):

In this study Andreev D and colleagues report that eosinophils promote bone density via inhibiting osteoclast formation and activity via one of its granule proteins, eosinophil peroxidase, using a variety of assays including a murine models of osteoporosis and serum transfer arthritis, ex vivo culture studies, histopathology and imaging of trabecular bone. Hyper-eosinophilia through the overexpression of IL-5 rescued this phenotype of increased pathology bone loss in the ovariectomy model. During the serum transfer arthritis model, the addition of two doses of EPX rescued WT mice from severe inflammatory bone loss suggesting that eosinophil-derived EPX may be the soluble factor inhibiting osteoclast activity in mice. Human data takes the form of synovial tissue biopsies, peripheral blood transcriptomics and blood eosinophil counts in rheumatoid arthritis patients versus healthy volunteers, correlating them with bone mass as measured by mCT.

If true, these findings would have significant medical implications because anti-eosinophil therapies are routinely being used for treatment of a variety of chronic conditions including asthma, EGPA, hypereosinophilic syndromes and EGIDs, with the hope of avoiding the toxic side effects of conventional therapies such as corticosteroids. While the data appear compelling, there are a number of major concerns which makes the results and conclusions less convincing.

We thank Reviewer 1 for the thoughtful remarks and suggestions. Below, you will find our responses addressing the raised comments.

Major Concerns (in no particular order)

1. It is unclear the relevance of measuring peripheral blood derived mRNA of osteoclast genes in human subjects— shouldn't these gene transcripts be in the bone only?

We thank the reviewer for this comment. In fact, monocytic precursors of osteoclasts enter the bone through the bloodstream and typically exhibit the expression of various genes associated with osteoclasts (Jacome-Galarza et al., 2019). As a result, analyzing alterations in the osteoclast population in patients' blood becomes relevant. We have addressed this matter on page 11 in the manuscript.

2. Details of concurrent or prior medications are not described for the patients. Steroids, methotrexate, are common drugs used to treat RA and have effects on numbers of eosinophils. Those who have mild RA may be on fewer medications or not be on meds at all. This would create a confounding element to the correlations reported in Figure 7. In fact, since long term corticosteroids use would decrease bone mineral density (and reduce eosinophil counts in the blood), this would be a significant confounder that would need to be accounted for.

We agree with the reviewer and have therefore entirely revised these analyses. We have now focused on the inclusion of early RA patients receiving methotrexate therapy. This allows to analyze a more homogeneous population and to limit the influence of treatment effects. No other conventional synthetic, targeted synthetic or biologic disease modifying anti-rheumatic drug (DMARD) were allowed to avoid treatment heterogeneity. Furthermore, RA patients receiving additional glucocorticoid treatment were separately analyzed from RA patients receiving no glucocorticoid treatment. We also improved the depiction of the results and show RA patients without glucocorticoids, RA patients with glucocorticoids and healthy controls within one graph for each of the parameters (cortical and trabecular bone; eosinophils and

Point-by-Point Reply

eosinophilic cationic protein (ECP) levels). We have incorporated these results into the updated Figure 7c-f and the manuscript on page 11 and 13.

The reviewer was right in suggesting that glucocorticoids may influence the correlation between eosinophils and bone. Hence, the link between bone mass and eosinophils and ECP was only found when no glucocorticoids were taken. If present, glucocorticoids suppressed eosinophils and ECP and the link to bone was lost.

3. Why was RA chosen to be studied? It is not typically thought to be an eosinophil-mediated disorder. Why not directly study patients with osteoporosis?

RA was chosen as this disease is one of the most important triggers for osteoporosis and fracture and since it has been shown that RA is also controlled by eosinophils (Andreev et al., Ann Rheum Dis. 2021; 80:451-468.). In accordance, our human study confirmed that both trabecular and cortical bone mass were lower in RA patients compared to healthy controls (updated Figure 7c-f and manuscript on page 11).

4. Age and gender are significant variables in bone mineral density and have an interaction too. A more complex statistical analysis should be done than just comparing mean ages.

The reviewer is correct. Demographic characteristics (age, sex and body mass index), disease-specific characteristics (disease duration and DAS 28 score), bone parameters (trabecular and cortical bone volume), and eosinophil parameters (eosinophil counts and ECP levels) of healthy controls and RA patients were recorded and are now included in the manuscript on page 29 in the new table 1. RA patients with (n=25) and without (n=25) glucocorticoids and healthy controls (n=25) were well matched with respect to age, sex and body mass index. RA patients without and with glucocorticoids had an overall short disease duration (1.5 and 1.6 years) and low disease activity (DAS28 score 3.18 and 3.26 units). Trabecular and cortical bone mass were lower in RA patients than in healthy controls. Trabecular but not cortical bone mass was significantly correlated to eosinophil counts and ECP levels in RA patients without glucocorticoids ($r=0.71$, $p=0.0001$; $r=0.55$, $p=0.0039$) and in healthy controls ($r=0.57$, $p=0.0025$; $r=0.57$, $p=0.0027$). No such correlation was found in RA patients exposed to glucocorticoids ($r=0.21$, $p=0.31$; $r=0.07$, $p=0.73$) (updated Figure 7c-f and manuscript on page 11).

5. Is mCT acceptable for measuring bone mineral density in human? Why not obtain DEXA scan results, which is typically what clinicians use?

HR-pQCT is the most advanced technique to analyze human bone in a standardized setting. It allows to separately measure trabecular and cortical bone with high accuracy. We have included this point in the manuscript on page 11.

6. mCT data of WT or untreated murine controls seem to be different from experiment to experiment, with the spread being as wide as some of the significant differences between treatment and no treatment. Please comment on the expected degree and range of bone loss in these models.

We acknowledge that the distinction between WT and control mice in this manuscript may be confusing. To clarify, WT mice are BALB/c controls for the eosinophil-deficient $\Delta dbiGATA$ mice. Meanwhile, IL-5tg/4get mice are compared to their 4get littermates, named control mice. To make it clearer, we have modified the description of the 4get mice in the manuscript.

In a previous study (Andreev et al., 2019), we demonstrated that the serum transfer arthritis (STA) model exhibits bone loss of approximately 27% (BV/TV of control: $8.81\% \pm 2.91$, BV/TV of STA: $6.47\% \pm 1.75$, $n=14$). Additionally, for the ovariectomy (OVX) model in BALB/c mice, bone loss ranging from 11% to 25% was reported (Roberts et al., 2019). These observed levels align well with the measurements obtained in our current study (STA: 11.4-31% bone loss, OVX: 15.4-27.1% bone loss). Moreover, it's worth noting that the STA and OVX models end at different time points and ages (11-12 weeks and 18 weeks, respectively). As a result, the bone mineral density can vary among these disease models.

7. Purifying eosinophils using positive selection via Siglec-F - these isolated eosinophils would be expected to die and extrude all sorts of non-specific proteins in this process, which could inhibit any cell growth. Would be useful to repeat with another cell type (neutrophils?) and see if their supernatants do the same – since PMNs don't make EPX. Alternatively, a blocking antibody to EPX would be a useful control to assess specificity as suggested. Or use negative selection to purify eosinophils.

We thank the reviewer for this comment. Therefore, we have verified the viability of eosinophils, maintaining levels of approximately 70-80% following a 48-hour co-culture with BMMs (updated Supplementary Figure 6c). Additionally, analyzing the secretome profile of these ex vivo cultured eosinophils revealed no upregulation of death-associated factors such as RAGE (Figure 3a and b).

As suggested by the reviewer, we have compared the impact of eosinophils and neutrophils on osteoclast differentiation. Accordingly, we isolated neutrophils via negative selection from the bone marrow of female BALB/c WT mice (9-10 weeks old) and co-cultured them with BMMs from female BALB/c WT mice (9-10 weeks old). Our findings demonstrate that neutrophils, in contrast to eosinophils, did not alter the number of osteoclasts or the expression of osteoclast-relevant genes. We have incorporated these results into the new Supplementary Figure 7a-e and the manuscript on page 5.

In addition, as recommended by the reviewer, we have inhibited the enzymatic activity of EPX using the compound 4-ABAH. Notably, EPX inhibition partially abolished the anti-osteoclastogenic effect of eosinophil supernatant (Figure 3g and h).

8. Seeing eosinophil related genes in the bulk-RNA seq data set is concerning for residual eosinophil contamination in the bulk RNA from these osteoclast cultures. Perhaps those eosinophil genes can be masked/removed from analysis, and the PCA be repeated to see if grouping still occurs.

Although the genes related to eosinophils show significant differential expression, we were able to identify only seven genes that are specifically associated with eosinophils (Alox15, Alox5ap, Ear1, Il5, Mpo, Retnlg, SiglecF). However, when studying osteoclasts stimulated with eosinophils, we observed over 800 differentially expressed genes (DEGs). We demonstrated that by excluding these eosinophil-specific genes from the bulk RNA sequencing analysis, the principal component analysis (PCA) still indicates clear differences between the control group and osteoclasts stimulated with eosinophils or eosinophil supernatant. However, the distinction between the two stimulated groups equalized. Considering the possibility that osteoclasts might also have internalized eosinophils, leading to the detection of eosinophil-associated genes in these cells, we decided to include all the DEGs detected in our analysis, and show the PCA after removing eosinophil-related genes in the updated Supplementary Figure 8 (on page 6-7 in the manuscript).

9. Since mCT data of trabecular bone is the main rodent clinical symptom, details on the following should be addressed:

9.1. Where on the tibia is the imaging acquired? Is a uniform size/volume acquired per sample? Best practices seem to image both axial and appendicular skeleton.

For the evaluation of the trabecular and cortex structure of the proximal tibia metaphysis, the volume of interest was determined starting at 420 μm (50 slices) from the middle of the growth plate and extending 1680 μm (200 slices) distally. These details have been included in the materials and methods section on page 16.

Unlike the well-documented bone decline observed in the axial skeleton for the OVX model, the extent of bone decline in the axial skeleton for the K/BxN serum transfer arthritis model is not extensively studied. To maintain uniformity in our analysis, we directed our focus towards assessing bone decline specifically in the appendicular skeleton. This has also been added on page 9 in the manuscript.

9.2. Comment on the choice of age of mice in this study. Given peak bone mineral density is at 16 weeks, trabecular bone decreases at weaning while cortical bone volume can continue to increase up to 7-12 months of age.

We opted to examine the bones of mice in the young adult phase, which typically ranges from 2-6 months. Since the bone loss models have varying durations (12 days for STA and 42 days for OVX), we were unable to precisely match the age of mice at the time of analysis. Nevertheless, all of our analyses were conducted during the young adult phase of mice. As stated in the materials and methods (page 14-15), steady state mice were assessed between 10-12 weeks of age, STA mice were evaluated at 11-12 weeks of age, and OVX mice were analyzed at 18 weeks of age.

9.3. Some mouse strains have lower trabecular volume specifically -e.g. C57BL/6 and specifically in the tibia – was this accounted for in the analysis between different mouse backgrounds?

The reviewer is correct in noting that the mouse background is crucial for the interpretation of bone parameters. Hence, we took care to ensure that all mouse lines used in the bone study shared the same background. As stated in the materials and methods on page 14, all of these mice were bred on a BALB/c background for more than 10 generation.

9.4. How do these magnitude of changes compare to other models of osteoporosis in mice?

The extent of bone loss observed in our study (STA: 11.4-31% bone loss, OVX: 15.4-27.1% bone loss) falls within the range of findings reported in previous studies using the STA and OVX models (Andreev et al., 2019; Roberts et al., 2019).

In comparison to the STA model, collagen-induced arthritis (CIA) has been shown to induce a higher degree of inflammatory bone loss (Pietschmann et al., 2022). However, we could not use this model, because BALB/c mice are resistant to CIA.

OVX represents the most common model for studying non-inflammatory bone loss, and it closely resembles postmenopausal osteoporosis, which is the primary type of osteoporosis in humans. In an investigation carried out by Jochems et al. in 2005, it was observed that both OVX and CIA caused a comparable reduction in bone mass among DBA/1 mice. Interestingly,

when these models were combined, their detrimental impact on bone loss was found to be additive.

10. Figure 1: Were male or female mice used for these experiments?

We used female and male mice that were age-matched for steady state analysis as the BV/TV in the proximal tibia is comparable in male and female BALB/c mice in the young adult phase (2-6 month)(Willingham et al., 2010). We also used age-matched female and male mice for STA analyses, because female and male K/BxN mice display similar severity of erosive arthritis (Gonçalves Dos Santos et al., 2020). For the sham and OVX experiments, we used female mice. This information is provided in the materials and methods section on page 14-15.

11. Figure 1 D and E: Δ dblGATA mice show significantly decreased bone mass compared to wild type controls. How do the authors reconcile this observation with published data by Kacena et al. in the Journal of Bone and Mineral Research that show increase in trabecular bone volume in GATA-1 deficient mice. Does eosinophil reconstitution of Δ dblGATA mice restore bone homeostasis?

The GATA-1 transcription factor is essential for the differentiation of various hematopoietic lineages, including eosinophils, mast cells, erythrocytes, and platelets (Meinders et al., 2016). In the study by Kacena et al., the GATA-1 knockout mice were generated differently and from a C57BL/6 background, leading primarily to the depletion of GATA-1 in megakaryocytes (Villeval et al., 1997), with no mention of eosinophils. In contrast, the Δ dblGATA mice used in this study have a selective loss of eosinophils due to the deletion of the high-affinity GATA-binding site in the GATA-1 promoter, while the development of other GATA-1-expressing lineages remains unaffected or subtly perturbed (Yu et al., 2002). Therefore, the bone phenotype of Δ dblGATA mice cannot be compared to that of GATA-1 knockout mice. We have discussed this point on page 12-13 in the manuscript.

Nevertheless, we quantified all GATA-1-expressing lineages, including granulocytes, mast cells, hematopoietic precursors, and the erythroid and megakaryocyte compartment in the bone marrow of Δ dblGATA mice as compared to WT controls. Our analysis showed a complete loss of eosinophils and only modest alterations in the basophil, erythroid and megakaryocyte compartment, while all other GATA-1-expressing lineages were unaffected. We have included these findings in the new Supplementary Figure 2 and in the manuscript on page 4.

Following the reviewer's suggestion, we have investigated whether the reconstitution of eosinophils in Δ dblGATA mice could restore their bone homeostasis. The bone mass could be partially restored in Δ dblGATA mice two weeks after reconstitution with 3×10^6 eosinophils by intravenous injection. Notably, the number of osteoclasts was significantly reduced in the tibial bone upon eosinophil reconstitution as compared with untreated Δ dblGATA mice. We have included these findings in the updated Supplementary Figure 3d-g and in the manuscript on page 4.

12. The IL-5 transgenic mouse expresses supraphysiological levels of IL-5 which not only affects the number of blood eosinophils in the mouse, but likely other cells that have the IL-5R in mice (e.g. basophils, mast cells perhaps, and certain B cell subsets). If the experiments are not done to address whether it is an eosinophil specific effect that leads to increased bone mineral density (e.g. crossing the IL-5tg with an eosinophil knockout to show that the protective effect is abrogated to show that it is eosinophil specific and not supported by IL-5 on non-

eosinophil cells), there needs to be some discussion of this point. Lastly, it should be noted that human B cells do not express the IL-5R which makes murine models utilizing IL-5 transgenes a little more challenging to compare directly with people.

We thank the reviewer for bringing this matter to our attention. In the original study conducted by Lee et al., 1997, the distribution of different immune cell types within IL-5tg mice was characterized. With the exception of the bone marrow, the overall count of all immune cells in the investigated tissues showed a significant increase, with the most notable changes observed in the total number of eosinophils. However, when analyzing the percentage distribution of each cell type, the majority of immune cells exhibited a reduction, except for eosinophils, which accounted for approximately 50-60% of all cells.

Due to variations in the background and age of mice used in the study of Lee compared to the current study, we made the decision to conduct a comprehensive analysis of all IL-5R α expressing cells in both the bone marrow and blood of IL-5tg/4get mice, in comparison to 4get controls (updated Supplementary Figure 4 and manuscript on page 4). In the bone marrow, we observed a slight increase in the populations of neutrophils and mast cells, while the different B-cell subsets were reduced. Nevertheless, the most significant alteration was observed in the eosinophil compartment, which displayed a 40-fold increase. In the blood, there was a decrease in the neutrophil and basophil fraction, while the number of mast cells and particularly eosinophils (15-fold) exhibited an increase. These findings indicate that the primary immune cells influenced by the consecutive expression of IL-5 are indeed eosinophils.

In response to the reviewer's suggestion, we attempted to cross IL-5tg/4get mice with Δ dblGATA mice. Unfortunately, despite our efforts over a period of more than six months, we were unable to obtain suitable offsprings for analysis. As an alternative approach, we treated Δ dblGATA mice with IL-5 to investigate whether IL-5 has eosinophil-independent effects on bone homeostasis. As indicated in the updated Supplementary Figure 3h-k, the intraperitoneal injection of 500ng of recombinant murine IL-5 daily for two consecutive weeks had no discernible effect on the bone and osteoclast phenotype of Δ dblGATA mice. Based on these findings, we suggest that eosinophils, rather than an elevated level of IL-5, are responsible for regulating osteoclast-mediated bone remodeling. Detailed information regarding these findings can be found in the manuscript on page 5.

13. Figure 2 H: Any LSFM imaging to support the statement in the text that "OC's likely internalize the Eos"

Unfortunately, LSFM can only visualize cell surface proteins through the use of fluorophore-labeled antibodies, as the staining process is done before mouse sacrifice. As a result, this technique cannot be used to visualize eosinophils that have been internalized by osteoclasts.

14. Figure 3 A and B: LIF and TGF β both are produced by eosinophils and have important roles in bone homeostasis. Was TGF β protein measured?

The reviewer is right that eosinophils are able to secrete LIF as well as TGF- β , which both have strong anti-osteoclastogenic properties. The proteome profiler array only showed low levels of LIF. For that reason, we focused our analysis on the highly expressed proteins MPO and EPX (Figure 3a and b). As TGF- β was not included in the proteome profiler array, we compared the levels of TGF- β in the supernatant of eosinophils with those of BMMs. The levels of TGF- β produced by unstimulated eosinophils were found to be even lower than those secreted by BMMs.

Point-by-Point Reply

Furthermore, we performed additional in vitro experiments where we employed a TGF- β inhibitor during the co-culture of BMMs with eosinophils and eosinophil supernatant. The inhibition of TGF- β signaling did not impact the anti-osteoclastogenic effect of eosinophils.

Detailed information regarding these findings can be found in the updated Supplementary Figure 9c-f and in the manuscript on page 8.

Reviewer #2 (Remarks to the Author):

Andreev and colleagues have done a tremendous effort to collect all the data presented in this manuscript. I am impressed and I would like to congratulate the authors on a well-written, comprehensive and complex manuscript - but admittedly also complicated dataset, which I feel is not always as clear as the authors state in the manuscript. This may simply be because I have overlooked something or simply did not understand it correctly. In this case, I would appreciate the authors if they could enlighten me, but maybe also consider if a potential future reader could have the same problems or thoughts. Therefore, I hope the authors will see my review in the right spirit since I am positive towards the manuscript although I am also skeptic.

To sum up my skepticism it is foremost based on what I see as a potential conflict between the in vivo and in vitro data. Although the in vivo effects are significant they are also modest in all experimental setups (when you look at the graphs). While the in vitro effects are very strong. Some of the in vitro effects I am also not totally convinced are direct effects of Eos. Below I will list a series of comments and requests. They are hard to group, but I hope the authors will find their way and recognize what my concerns are:

We thank Reviewer 2 for showing interest in our research and for providing valuable suggestions to improve our manuscript. Please find our responses to the comments raised below.

- Throughout the entire manuscript please modify statements that are overstatements such as: Do the Eos really CONTROL OCs in vivo. The differences are not convincing enough for such a strong statement.

We thank the reviewer for bringing this to our attention. We have modified the statement saying that eosinophils influence steady-state bone remodeling (page 3).

- Page 6 bottom: does the use of an inhibitor prove anything? Or does it maybe suggest? No inhibitor is 100% effective or specific...

We thank the reviewer for this comment. We have made the necessary changes to this paragraph to address this matter (page 8).

- Page 8 title: do Eos really protect? No - maybe reduce but not protect!

We have revised the title as follows: High eosinophil numbers reduce postmenopausal and inflammatory bone loss (page 10).

- Page 9 center: verify – no the results do not verify and they certainly do not show that Eos control human bone homeostasis. What is shown are correlations – no more and no less.

We have revised this paragraph accordingly (page 11).

- Page 2 bottom: Is TRAP and MMP9 crucial for osteoclastic bone resorption?

As indicated on page 2, we have changed this statement, saying that these enzymes are necessary for proper bone resorption (page 2).

- There are many such exaggerations – please correct and please be thorough!

We appreciate the reviewer for providing this comment. We have made the necessary changes to the manuscript to address this issue.

- What is the purity of the Eos cultures? 86%? And how about after 48h culture? I feel the authors do not consider well enough the potential consequences of contaminating cells.

Point-by-Point Reply

The reviewer's assumption regarding the approximate purity of eosinophils being 90% is correct. We have included this information in the updated Supplementary Figure 6a and b. Additionally, we performed an evaluation of eosinophil purity after 48 hours of culture, and the results demonstrate that the purity remains above 90%.

- Is the FBS used for Eos and OCs the same?

Indeed, the same FBS was used in all in vitro experiments including the single culture of BMMs, the co-culture of eosinophils with BMMs, and the single culture of eosinophils to generate the eosinophil supernatant. This information has been included in the material and methods section on page 19.

- In cell cultures how are the control conditions treated? What would be a suitable control for e.g. the co-culture experiment? I find that is hard to answer, but it is critical to interpret the result correctly.

We extend our appreciation to the reviewer for this valuable comment. We made every effort to ensure the accuracy of the control condition. Specifically, we maintained the same FBS and identical concentrations of M-CSF and RANKL in the control condition as in the eosinophil supernatant.

Nonetheless, we took this comment seriously and have added further controls, such as treating BMMs with neutrophils and BMM supernatant. Our new results indicate that unlike eosinophils, neutrophils did not exhibit any discernible influence on the number of osteoclasts or the expression of osteoclast-relevant genes. We have included these outcomes in the updated Supplementary Figure 7a-e and on page 5 of the manuscript. Moreover, the introduction of BMM supernatant exhibited no influence on the osteoclast differentiation potential, in contrast to the osteoclast-inhibitory effects observed with eosinophil supernatant. These results can be found in the updated Supplementary Figure 7f-h, as well as on page 6 of the manuscript.

- Regarding the co-culture effects, I am sceptic that the effects observed are not just a density problem. The preOCs may simply not find each other? Would that not be observed when co-culturing with many other cell type?

To exclude the possibility of density-related issues in the direct culture of BMMs with eosinophils, we conducted transwell experiments. In these experiments, eosinophils and BMMs were separated by a 1 µm pore size membrane, enabling soluble protein exchange while preventing cell contact. Remarkably, similar to the direct co-culture, eosinophils significantly inhibited osteoclast differentiation in the transwell experiment, despite the spatial separation. We have included these findings in the updated Supplementary Figure 6i and j and on page 6 of the manuscript.

Furthermore, when BMMs were co-cultured with an equivalent number of neutrophils, osteoclast differentiation remained unaffected. In contrast, the presence of an equal number of eosinophils impaired the differentiation of osteoclasts (Supplementary Figure 7a-e and on page 5 of the manuscript.)

- In conditioned media experiments what is added to the control cells? Is fresh media added here or was this also incubated for 48h prior to addition?

We ensured that the control conditioned media and eosinophil supernatant had the same FBS, and the same M-CSF and RANKL concentrations. In order to compare the effects of eosinophil

supernatant with a control cell supernatant, we utilized the supernatant from BMMs that had been cultured for 48 hours. Unlike the eosinophil supernatant, the addition of the control supernatant obtained from BMMs had no discernible impact on osteoclast differentiation or the expression of osteoclast-associated genes. These findings are presented in the updated Supplementary Figure 7f-h, as well as on page 6 of the manuscript.

- Quantifications of OCs are consistently shown as more than or equal to 5 nuclei. This needs to be more than or equal to 3 nuclei.

We acknowledge that when quantifying primary human osteoclasts, it is appropriate to count cells with three or more nuclei. However, due to the higher density and nuclearity of murine primary osteoclasts compared to human osteoclasts, we have found it more accurate to count cells with five or more nuclei for the murine samples.

Nevertheless, as suggested by the reviewer, we have added the quantification of osteoclasts with three or more nuclei after supplementation with eosinophils or eosinophil supernatant (updated Figure 2a and c and manuscript on page 5 and 6). The results of this reanalysis indicate that even though a greater number of osteoclasts are counted overall, the observed effects of supplementation with eosinophils or eosinophil supernatant remain consistent and unchanged.

By conducting this additional analysis, we demonstrate that our conclusions and interpretations are not affected by the nuclei counting criteria used for murine osteoclasts.

- Fig. 1h, 1n and SF2: Isn't it puzzling that Acp5 increases dramatically and Ctsk is unchanged in Fig. 1h? Maybe this explains the modest in vivo effects on bone? In Fig. 1n Acp5 expression is lost but Ctsk is not significantly reduced? SF2 here the in vivo bone effect seems modest, but Ctsk is sky high? Does this add up and does this support the title and conclusions throughout the manuscript?

The reviewer's observation is accurate that the steady state effects on bone in the Δ dbpGATA and IL-5tg/4get mice may be considered modest, albeit significant. However, it is important to highlight that the impact on in vivo osteoclast number is notably evident and is mirrored by the expression profile of osteoclast-associated genes. These changes extend beyond just Ctsk and Acp5 gene expressions and also encompass other genes related to osteoclast differentiation and function, such as Nfatc1 and Mmp9. As a result, we can confidently assert that the overall gene expression pattern reflects the bone phenotype. While the steady state effect of eosinophils on bone metabolism may appear modest, it expands in the context of inflammatory arthritis and oestrogen-deficiency-induced osteoporosis. We have discussed this point on page 11 in the discussion.

- The authors chose not to use bone as a substrate to test the activity of OCs in vitro. Therefore, the authors can also not make any comparison to in vivo effects since mouse bone does not just consist of hydroxyapatite but also of large amounts of collagen – and this is a critical difference. I would strongly urge the authors to use a bone substrate! The other substrate is convenient to use – but not even close to the natural substrate.

We thank the reviewer for this suggestion. In response to this, we conducted further experiments by culturing BMMs in the absence or presence of eosinophils and eosinophil supernatant on bone slices derived from the cortical part of bovine femurs. Detailed quantifications and representative images are now provided in the updated Figure 3 l and m

and described in the manuscript on page 7. Consistent with the findings on synthetic surfaces, we observed a decrease in bone resorption of femoral bone when BMMs were stimulated with both eosinophils and eosinophil supernatant.

- How were the images chosen to use for quantification of mineral resorption? And how many pictures from each well?

The whole well was imaged with Keyence, stitched together and quantified with Image J. This information has been included in the materials and methods section on page 21.

- Does dbIGATA only affect the presence of Eos? How about other cell types? Are other cells in the bone marrow affected directly or indirectly? Could this potentially affect interpretations of the data?

The transcription factor GATA-1 is crucial for the differentiation of several hematopoietic lineages, such as erythrocytes, platelets, mast cells, and eosinophils (Meinders et al., 2016). However, in the Δ dbIGATA mice used in this study, only the high-affinity GATA-binding site in the GATA-1 promoter is deleted, resulting in the selective loss of eosinophils, while the development of other GATA-1-expressing lineages (such as erythroid, megakaryocytic, mast) remains unaffected or only slightly affected (Yu et al., 2002).

Nonetheless, to address the reviewer's inquiry, we examined all GATA-1-expressing lineages, including granulocytes, mast cells, hematopoietic precursors, and the erythroid and megakaryocyte compartment in the bone marrow of Δ dbIGATA mice in comparison to WT controls. Our analysis revealed that Δ dbIGATA mice have a complete loss of eosinophils and only modest alterations in the basophil, erythroid and megakaryocyte compartment, while all other GATA-1-expressing lineages remain unaltered. These findings have been incorporated into the new Supplementary Figure 2 and the manuscript on page 4. Hence, the bone phenotype observed in Δ dbIGATA mice is most likely attributable to the specific loss of the eosinophil lineage in the bone marrow compartment.

- Do not show linear correlation lines when doing Spearman analyses! I assume you have chosen a Spearman correlation because there is no linear correlation. Especially in F7e and f they are far from linear – actually the correlations are not very convincing although significant.

As suggested by the reviewer, we have removed the linear correlation lines in the human gene expression correlations (updated Figure 7b) and in the revised human correlations between eosinophils, eosinophilic cationic protein (ECP) and bone mass (updated Figure 7c-f). In addition, we have done a more thorough analysis of the human data: We focused on early rheumatoid arthritis patients that are treated with methotrexate to avoid any bias from treatments. Furthermore, we divided the RA patients into two groups: one without any supplementary glucocorticoid treatment (only methotrexate) and those receiving additional glucocorticoids. We also carefully balanced the groups (including healthy controls) for age, sex and body mass index. When no glucocorticoids were present, the correlation between bone mass and eosinophils, as well as ECP, was highly significant. However, in patients receiving glucocorticoids, eosinophils and ECP were suppressed, resulting in the loss of correlation to bone mass. These newly analyzed datasets emphasize the potential involvement of eosinophils in bone loss associated with arthritis in human subjects and are discussed in the manuscript on page 11 and 13.

- Fig. 1 – what does it mean for cells to be close in the bone marrow? The bone marrow cavity of a mouse is tiny. It is essential that the authors measure the distances between Eos and OCs in sections when coming with such a statement – images do not support this statement alone. How about direct contacts? Also all the in vitro results only investigate effects on OCs differentiation but not on mature OCs – so is it really relevant whether they are close to mature OCs?

The images demonstrate the presence of eosinophils in both the metaphysis and diaphysis of long bones, suggesting potential interactions with both osteoclast progenitors and mature osteoclasts. We have revised the statement on page 4 accordingly.

To investigate whether direct contact between eosinophils and osteoclasts occurs, we have added an additional image in Figure 1b, which depicts a direct interaction between these cells.

*Furthermore, we conducted an in vitro experiment, where we treated more mature osteoclasts with eosinophils and eosinophil supernatant, starting on day 2 of differentiation. Interestingly, direct co-culture with eosinophils did not affect the late stage of osteoclast differentiation. However, the supernatant from eosinophils partially decreased the number of osteoclasts and significantly reduced the expression of the bone resorption-related genes *Acp5* and *Ctsk*. These findings have been included in the updated Supplementary Figure 6f-h and the manuscript on page 5 and 6.*

- How could the authors show that it is truly a soluble factor from Eos that cause an effect on preOCs? Is the use of conditioned media enough? In this context it is especially important to consider how the control is treated?! And from what I can read in the methods, I am uncertain about this.

Following the reviewer's suggestion, we conducted transwell experiments to demonstrate that eosinophils effectively inhibit osteoclast formation even when separated from BMMs by a 1 μ m membrane. These experiments suggest that the primary cause is likely the release of soluble factors from eosinophils. We have included these results in the updated Supplementary Figure 6i and j and on page 6 of the manuscript.

- Page 5 bottom: the actin rings in Fig. 2i are not convincing as having anything to do with active resorption – the pictures are simply not useful. I would also like to remind the authors that they have chosen not to use bone so the OCs is simply sitting on glass where such sealing zone structures does not look like on bone. Please revise carefully.

We agree with the reviewer, nevertheless, through our bulk RNA sequencing analysis, we observed downregulation of actin skeleton-associated genes in osteoclasts following their culture with eosinophils and eosinophil supernatant. Consequently, investigating the formation of actin-rich podosome belts, also known as sealing zones, in these osteoclasts became an important approach. To visualize the sealing zones in high resolution, it was necessary to culture BMMs on glass slides, which is considered the optimal method for this purpose. Moreover, to draw relevant conclusions about active resorption, we conducted experiments on both synthetic bone substrates and femoral bone slices (see Figure 3j-m).

- Statistics: Statistical tests are parametric tests despite that in some experiments n=4?! I insist that the authors are very critical towards which tests they use and when. Have the authors checked that all datasets meet the requirements for doing the tests chosen? From the statistics

Point-by-Point Reply

paragraph, it does not seem so. Please go carefully through all tests used for all data sets and make sure that there are statistically sound arguments to use the chosen test.

In response to the reviewer's suggestions, we conducted a thorough review of all statistical calculations. As indicated in the materials and methods section on page 24, we determined statistical significance using appropriate methods for different scenarios. Specifically, for single comparisons involving normally distributed variables, we employed the unpaired two-tailed t-test, whereas for comparisons with non-normal variables, we utilized the Mann-Whitney test.

To address multiple comparisons, we applied parametric tests such as one-way ANOVA or two-way ANOVA when dealing with normally distributed variables. For cases involving non-normal variables, we employed the Kruskal-Wallis test, which is a non-parametric alternative.

By employing these rigorous statistical analyses, we aimed to ensure the reliability and accuracy of our findings.

REVIEWER COMMENTS

Reviewer #2 (Remarks to the Author):

I would like to congratulate the authors on a very thorough response to both Reviewer1 and 2 and it is my impression that this has substantially improved this manuscript.

At this stage, I only have three issues remaining, but which I insist on:

- I appreciate that the authors have made the efforts of trying to do a bone resorption assay on bone slices. However, this does not do any good for the manuscript - at all. Bone resorption levels of 0.3% and below is by far not convincing and when looking at the included images in Fig. 2m I am definitely not convinced! I cannot see a single convincing bone resorption cavity in these images! So, I am not in any way convinced that assay has been correctly and analyzed correctly. This pulls down the quality of an otherwise convincing manuscript. I ask the authors to REMOVE Figs. 2l and M again and all related text that was included in this regard. Instead, the authors in the manuscript should be cautious not to talk about Fig. 2J as assessment of bone resorption, but only refer to the activity measured as "demineralization activity" - it is also not artificial bone - it is mineral/hydroxyapatite so please only use this terminology.

From the authors analyses it also becomes evident that the effects observed are due to effect on early OC differentiation and not directly on the mature OCs. So it is OK to downscale this part of the story.

- I also do not find that the images of the podosome belts does anything good for the manuscript. If the authors wish to keep these images this is OK, but do NOT refer to them as sealing zones that has anything to do with active bone resorption. I appreciate the argumentation the authors have included in response to my question. Nevertheless, the major difference between a podosome belt on glass and a sealing zone on bone is its micro anatomical structure and not so much its individual components. I am aware that podosome belts are often referred to in the literature as "sealing zones", but just because a misunderstanding is repeated, this does not make it more correct. So please make sure NOT to refer to this as a "sealing zone", but rather as a podosome belt, and do NOT make a direct link between this observation and bone resorption. Therefore, I would also recommend to simply remove them, they do not add important information.

- Regarding the statistics I am still not satisfied. In many instances parametric statistics are still used on datasets with e.g. $n=4$ or where variances are clearly not matching. At this stage I would like the Editor to involve a statistician on behalf of the journal to ensure that the data presented in this manuscript are analyzed in the most proper way and then help the authors in making the necessary changes.

Congratulations on a thorough study that I will support for publication when the above-mentioned adjustments have been done.

Reviewer #3 (Remarks to the Author):

I. want to commend the authors on the thorough and thoughtful response to reviewers' comments. The manuscript presents compelling findings that have significant medical implications.

REVIEWER COMMENTS

Reviewer #2 (Remarks to the Author):

I would like to congratulate the authors on a very thorough response to both Reviewer1 and 2 and it is my impression that this has substantially improved this manuscript. At this stage, I only have three issues remaining, but which I insist on.

We thank Reviewer 2 for the favorable feedback. Below, you will find our responses addressing the remaining issues.

I appreciate that the authors have made the efforts of trying to do a bone resorption assay on bone slices. However, this does not do any good for the manuscript - at all. Bone resorption levels of 0.3% and below is by far not convincing and when looking at the included images in Fig. 2m I am definitely not convinced! I cannot see a single convincing bone resorption cavity in these images! So, I am not in any way convinced that assay has been correctly and analyzed correctly. This pulls down the quality of an otherwise convincing manuscript. I ask the authors to REMOVE Figs. 2l and M again and all related text that was included in this regard. Instead, the authors in the manuscript should be cautious not to talk about Fig. 2J as assessment of bone resorption, but only refer to the activity measured as "demineralization activity" - it is also not artificial bone - it is mineral/hydroxyapatite so please only use this terminology. From the authors analyses it also becomes evident that the effects observed are due to effect on early OC differentiation and not directly on the mature OCs. So it is OK to downscale this part of the story.

We thank the reviewer for providing a thorough assessment of the bone resorption assay. As recommended by the reviewer, we have excluded Figures 2l and m and implemented all necessary modifications in the manuscript. Additionally, we have substituted all references to "bone resorption activity" with "demineralization activity" and consistently employed the term "hydroxyapatite".

I also do not find that the images of the podosome belts does anything good for the manuscript. If the authors wish to keep these images this is OK, but do NOT refer to them as sealing zones that has anything to do with active bone resorption. I appreciate the argumentation the authors have included in response to my question. Nevertheless, the major difference between a podosome belt on glass and a sealing zone on bone is its micro anatomical structure and not so much its individual components. I am aware that podosome belts are often referred to in the literature as "sealing zones", but just because a misunderstanding is repeated, this does not make it more correct. So please make sure NOT to refer to this as a "sealing zone", but rather as a podosome belt, and do NOT make a direct link between this observation and bone resorption. Therefore, I would also recommend to simply remove them, they do not add important information.

We thank the reviewer for emphasizing the distinction between the podosome belt and the sealing zone. In light of this observation, we agree that the images do not provide valuable information. Consequently, we have removed the images and incorporated all required modifications in the manuscript.

Regarding the statistics I am still not satisfied. In many instances parametric statistics are still used on datasets with e.g. n=4 or where variances are clearly not matching. At this stage I

would like the Editor to involve a statistician on behalf of the journal to ensure that the data presented in this manuscript are analyzed in the most proper way and then help the authors in making the necessary changes.

Following the reviewer's suggestion, we consulted a statistician from our department, who conducted an additional review of the statistical analyses for our experimental datasets. All adjustments made in accordance with this advice are highlighted in the manuscript. Additionally, we have included an Excel file that encompasses the raw data, information on the conducted normality tests, summaries of the statistical comparisons and tests performed, as well as the associated p-values and significances. If considered necessary, the editor is welcome to engage an independent statistician for further consultation.

Reviewer #3 (Remarks to the Author):

I want to commend the authors on the thorough and thoughtful response to reviewers' comments. The manuscript presents compelling findings that have significant medical implications.

We thank Reviewer 3 for dedicating his/her time to assess our manuscript. We are very pleased about the positive comment.

REVIEWERS' COMMENTS

Reviewer #2 (Remarks to the Author):

The authors have done a substantial effort to improve their statistical analyses. Although Shapiro-Wilk testing is not optimal it is now so few of the many many results that may still be questioned - that I am now happy to accept.

Congratulations on an impressive study!